# Changes in Sea Surface Temperature and Sea Ice Concentration in the Arctic Ocean over the Past Two Decades

**Meng Yang** [1,2,3], **Yubao Qiu** [2,3,*], **Lin Huang** [2,3,4], **Maoce Cheng** [1,3], **Jianguo Chen** [1], **Bin Cheng** [5] **and Zhengxin Jiang** [2,3,4]

1. School of Earth Resources, China University of Geosciences, Wuhan 430074, China
2. International Research Center of Big Data for Sustainable Development Goals, Beijing 100049, China
3. Key Laboratory of Digital Earth Science, Aerospace Information Research Institute, Chinese Academy of Sciences, Beijing 100049, China
4. University of Chinese Academy of Sciences, Beijing 100049, China
5. Finnish Meteorological Institute, FI00101 Helsinki, Finland
* Correspondence: qiuyb@aircas.ac.cn

**Abstract:** With global warming, the decrease in sea ice creates favorable conditions for Arctic activities. Sea surface temperature (SST) is not only an important driven factor of sea ice concentration (SIC) changes but also an important medium of the ocean–atmosphere interaction. However, the response of sea surface temperature to Arctic sea ice varies in different sea areas. Using the optimal interpolated SST data from the National Centers for Environmental Information (NCEI) and SIC data from the University of Bremen, the temporal and spatial characteristics of SST and SIC in the Arctic above 60°N and their relationship are studied, and the melting and freezing time of sea ice are calculated, which is particularly important for the prediction of Arctic shipping and sea ice. The results show that (1) the highest and lowest monthly mean Arctic SST occur in August and March, respectively, while those of SIC are in March and September. The maximum trends of SST and SIC changes are in autumn, which are +0.01 °C/year and −0.45%/year, respectively. (2) There is a significant negative correlation between the Arctic SST and SIC with a correlation coefficient of −0.82. (3) The sea ice break-up occurs on Day of the Year (DoY) 143 and freeze-up occurs on DoY 296 in the Arctic. The melting and freezing processes lasted for 27 days and 14 days, respectively. (4) The Kara Sea showed the strongest trend of sea ice melting at −1.22 d/year, followed by the Laptev Sea at −1.17 d/year. The delay trend of sea ice freezing was the most significant in the Kara Sea +1.75 d/year, followed by the Laptev Sea +1.70 d/year. In the Arctic, the trend toward earlier melting of sea ice is smaller than the trend toward later freezing.

**Keywords:** Arctic; sea surface temperature; sea ice concentration; melting; freezing

## 1. Introduction

In recent decades, the temperature in the near-surface Arctic has warmed twice as fast as the global average [1–3]. This Arctic amplification phenomenon intensifies the melting of Arctic sea ice and has a great impact on global climate change. SST is a key factor affecting the melting of sea ice. In the early stage, SST was monitored by ships, measuring the sea temperature from tens of centimeters to below 5 m [4]. Later, through the calculation and analysis of buoy and satellite data, the temperature at each depth was obtained according to different sensors and methods. As a major ocean variable, SST affects the exchange of energy, momentum, and gas between the ocean and the atmosphere. As an important part of the climate system, SST is often used as a key input parameter in numerical weather prediction and ocean prediction systems [5–7], which is also of great significance in revealing climate change [8].

One of the major oceanic characteristics of the ocean includes its ability to move large amounts of heat, fresh water, carbon, and other properties over long distances, affecting

the ocean and the Earth's climate, locally to globally, and always at scale [9]. Global SST has been increasing at a rate of approximately 0.016 °C per year from 1993 to 2020. This equates to an increase of approximately 0.43 °C worldwide. Compared with low-latitude regions, the long-wave radiation temperature in high-latitude regions is enhanced, which strengthens the heat exchange between the atmosphere and the ocean, and the huge heat transfer accelerates the rise of SST. In the past 140 years, global SST has increased at a rate of 0.0038 °C/a [10]. The annual average SST of the Arctic Ocean during the period of 1982–2018 was 1.32 ± 1.5 °C, and the overall warming trend was approximately 0.036 ± 0.03 °C/year. The annual mean temperature in the Barents Sea ranges from 0.2 °C to 3 °C, with a wide variation trend of −0.01 °C to 0.05 °C/year. The variation range in the Greenland Sea is larger, at −0.03 °C/year to 0.02 °C/year. The Norwegian Sea has a warming trend of 0.04–0.07 °C/year, while other marginal seas show a relatively weak warming trend of −0.01–0.01 °C/year [11]. Influenced by latitude, seasonal regulation, and the nature of ocean currents, SST variation has obvious regional characteristics [12]. The variation of SST in different sea areas is not uniform.

The melting of ice in the Arctic may influence global climate, alter ecosystems, and create hazards, posing major challenges to nature and human activities [13]. The current sea-ice loss is driven by both atmospheric and oceanic processes [14]. Sea ice melting starts from the surface receiving atmospheric and solar heat fluxes, and solar radiation and ocean heat fluxes act together [15]. Seawater temperature is an important parameter in the estimation of ocean heat fluxes. In November 2018, the SIC and SST time series changes observed by satellite showed that the unusually warm SST in the Chukchi Sea delayed sea ice freezing, and the warm water prevented the advance of sea ice growth before the ocean heat was fully released into the atmosphere [16]. With the decrease in Arctic sea ice, the development of Arctic shipping ushered in opportunities. During 1978–2017, SST in the Northwest Passage showed an upward trend, and there was a significant negative correlation between SST and SIC. The correlation coefficient of SST and SIC in the northern Northwest Passage was as high as −0.96 [17]. Among the three Arctic shipping routes, the Northeast Shipping Route has the best navigable conditions [18–21] and is one of the important regions for energy exchange between land and sea in the Arctic region. Sea ice, the biggest obstacle to shipping in the Arctic, is also influenced by climatic factors such as storms, cyclones, and frontal activity [22–24]. From 1979 to 2018, the extent of sea ice in the Barents Sea and the Kara Sea showed downward trends of $-23 \pm 2.5\%$ decade$^{-1}$ and $-7.3 \pm 0.9\%$ decade$^{-1}$, respectively. The extent of sea ice in the Barents Sea in winter decreased more than that in the Kara Sea, while in summer, the sea ice cover in the Barents Sea has a downward trend of $-4.1 \pm 0.7 \times 10^3$ km$^2$yr$^{-1}$ due to the influx of the warm Atlantic current. At the interannual scale, the Barents Sea–Kara Sea SST is negatively correlated with SIC. In winter, SST is negatively correlated with SIC in the Barents Sea compared with the Kara Sea, which is correlated with heat loss from the ocean to the atmosphere. In summer, SST in the Barents Sea shows no significant negative correlation with SIC [25]. With the increase in the Arctic temperature, SST has attracted more attention. SST is an important medium of the atmosphere–ice–ocean interaction. Correlation analysis of SST and SIC can promote the understanding of the differences among different Arctic Sea areas.

SST and sea ice in the Arctic are changing rapidly. Significant temperature increases and sea ice loss are critical to environmental change in the Arctic and also affect Arctic shipping [26], and the melting and freezing of sea ice are closely related to this. Between 2011 and 2020, Arctic sea-ice thickness was 1.87 ± 0.10 m at the start of the melting season in May and 0.82 ± 0.11 m by the end of the melting season in August. The critical melt period is from May to September [27]. With the loss of sea ice, the Arctic melting season has been significantly prolonged, showing an earlier start time of melting and a later start time of freezing. Based on the satellite time series analysis from 1979 to 2018, the date of autumn sea ice freezing is significantly correlated with the extent of sea ice in early summer. Sea ice loss is the main reason for the delay of autumn freezing, which is accompanied by

the decrease in the surface albedo in summer, the increase in net shortwave radiation, and the increase in skin temperature [28]. The melting process of perennial Arctic sea ice is different from that of seasonal sea ice, and it is transforming into seasonal ice cover [29,30]. The perennial ice is usually near the North Pole, and the snow on its surface melts, forming large lakes, which carry heat and freshwater with the drainage, permeate along the flaws of the ice, and remain as melt ponds. Freshwater basal melt and solar warming of the ocean are ongoing, causing the SIC to slowly decline and basal melt to start, leaving the sea ice to persist throughout the summer. Seasonal sea ice generally forms at the edge of the Arctic. Under the influence of freshwater basal melt and solar warming of the ocean, the SIC decreased rapidly with snow remaining on the ice surface and basal melting. As the snow melts, small melt ponds form, the melt ponds drain to bring heat and fresh water to the sea ice, and the melt ponds become smaller until the ice melts completely and forms the open sea [31]. The area of rapid sea ice change is primarily in the thin ice area. The growth and melting of sea ice are affected by many factors, and Arctic sea ice is dynamic. It is very important for the Arctic climate system to determine the melting and freezing time of sea ice, especially in the sensitive navigation area, which is related to the passage of ships. In addition, changes in melting and freezing in various ocean areas contribute greatly to understanding the Arctic and global melt pools, albedo, and energy changes.

In this paper, the Arctic region above 60°N is divided into ten sea areas: The Norwegian Sea, Barents Sea, Kara Sea, Laptev Sea, East Siberian Sea, Chukchi Sea, Beaufort Sea, Northwest Passage, Baffin Bay, and Greenland Sea (Figure 1). The variation of SST and sea ice in each Arctic Sea area from 2002 to 2021 and the correlation between them are analyzed. Finally, the time nodes of melting and freezing of Arctic sea ice are calculated.

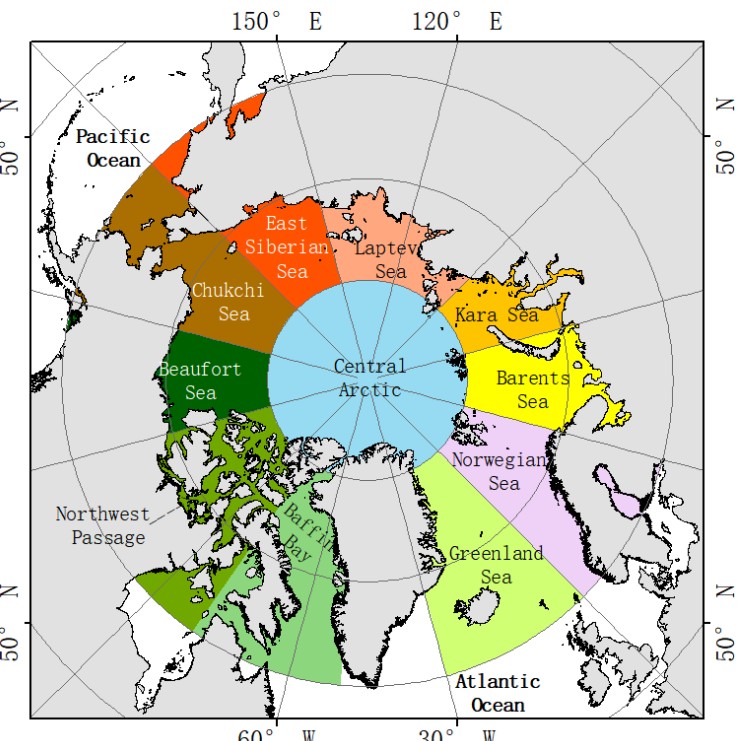

**Figure 1.** Location of Arctic marginal seas in the study area.

## 2. Data

### 2.1. Sea Surface Temperature

SST data from the National Center for Environmental Information (NCEI, https://www.ncei.noaa.gov/, accessed on 16 June 2022) daily Optimum Interpolation Sea Surface Temperature dataset DOISST Version 2.1, with a spatial resolution of 0.25° × 0.25°, has provided daily global SST data since September 1981 [32,33], which has been well applied

in the related studies of Arctic and Pacific SST [12,34–37]. The DOISST incorporates observations from different platforms (satellites, ships, buoys, and Argo floats) into a regular global grid. The bias of DOISST v2.1 is −0.07 °C in the global ocean. Compared with Argo observations, the bias is −0.04 °C. The difference compared to the Group for High-Resolution SST (GHRSST) Multiproduct Ensemble (GMPE) product is −0.01 °C [38]. OISST V2.1 adjusts and compensates for the error caused by sensor deviation and platform difference through multi-party data merging and interpolates the blank missing data to generate a complete global regular grid data, and the SST is in the unit of °C [38–40]. We used data from 2002 to 2021 to analyze changes in SST in the Arctic.

### 2.2. Sea Ice Concentration

SIC data are obtained from the daily Sea Ice density data of the University of Bremen (https://seaice.uni-bremen.de/data, accessed on 16 June 2022), which varies between 0 and 100%. The data products are generated by the ARTIST Sea Ice (ASI) algorithm. Daily SIC data from 2002 to the present are provided with a spatial resolution of 3.125 km × 3.125 km using a standard latitude and longitude grid for polar projections. This dataset utilized Advanced Microwave Scanning Radiometer for EOS (AMSR-E) class 1A data from NASA's Aqua satellite and Advanced Microwave from JAXA's GCOM-W1 satellite Scanning Radiometer 2 (AMSR2) level 1B data [41]. Daily SIC data from 2002 to 2021 were used to analyze the response relationship between SIC and SST in the Arctic. SIC data are missing from May 2002, May–June 2012, and November–December 2011 due to sensor failures. When the SIC is 100%, the absolute error is 5.7%, and the absolute error decreases with the increase in SIC [32].

### 2.3. Sea Ice Extent

The sea ice extent is derived from the U.S. National Ice Center (USNIC). USNIC has determined Arctic sea ice extent based on analysis using the USNIC's Interactive Multi-Sensor Snow and Ice Mapping System (IMS), used in the Multisensor Analyzed Sea Ice Extent (MASIE) product, jointly created with the National Snow and Ice Data Center (NSIDC). The data download site is https://usicecenter.gov/Products/ (accessed on 3 October 2022). The daily sea ice boundary product utilizes a variety of near-real-time satellite data, buoy data, meteorological data, and reanalysis of the current sea ice conditions. USNIC uses this 3-day running average to calculate the ice extent to account for short-term spatial variation in sea ice edge location and to reduce subjectivity and variability in determining ice extent. The data format we downloaded is SHP. We used statistics from NSIDC and NASA to determine the dates of the maximum and minimum Arctic sea ice extent and added them to the melting and freezing results (Table 1). Sea ice extent in the Arctic typically reaches a maximum in March and a minimum in September.

**Table 1.** Dates of Arctic sea ice maximum extent and minimum extents.

| Date | Maximum | Minimum |
|------|---------|---------|
| 2005 | 12 March | 20 September |
| 2006 | 12 March | 15 September * |
| 2007 | 12 March | 18 September |
| 2008 | 15 March * | 19 September |
| 2009 | 5 March | 13 September |
| 2010 | 31 March | 21 September |
| 2011 | 9 March | 11 September |
| 2012 | 15 March * | 17 September |
| 2013 | 28 February | 13 September |
| 2014 | 21 March | 17 September |
| 2015 | 25 February | 9 September |
| 2016 | 23 March | 10 September |
| 2017 | 7 March | 13 September |

**Table 1.** *Cont.*

| Date | Maximum | Minimum |
|------|---------|---------|
| 2018 | 17 March | 23 September |
| 2019 | 13 March | 18 September |
| 2020 | 5 March | 16 September |
| 2021 | 21 March | 16 September |
| 2022 | 25 March | 18 September |

* estimated.

## 3. Methods

### 3.1. Resampling Method

The annual, quarterly, and monthly average SST and SIC data are calculated by

$$\overline{X}_n = \frac{\sum_{i=1}^{n} x_i}{n} \tag{1}$$

$\overline{X}_n$ is the monthly average of the data, $n$ is the number of days per month, and $x_i$ is the daily value of the point. Using daily data, we added up the values of a position point in the month and divided it by the corresponding days to obtain the monthly average value of the position point. The daily SST and SIC of the current month were averaged to create the spatial distribution of the monthly average [16,42]. Similarly, annual data were the annual average, and quarterly data were the seasonal average. The spatiotemporal variations of Arctic SST and SIC were analyzed.

### 3.2. Linear Regression Analysis

The linear regression equation can be expressed as:

$$y = bx + a \tag{2}$$

$$b = \frac{\sum_{i=1}^{n}(x_i - \overline{x})(y_i - \overline{y})}{\sum_{i=1}^{n}(x_i - \overline{x})^2} = \frac{\sum_{i=1}^{n} x_i y_i - n\overline{xy}}{\sum_{i=1}^{n} x_i^2 - n\overline{x}^2}, a = \overline{y} - b\overline{x} \tag{3}$$

$$\overline{x} = \frac{\sum_{i=1}^{n} x_i}{n}, \overline{y} = \frac{\sum_{i=1}^{n} y_i}{n} \tag{4}$$

The independent variable $x$ is time, the dependent variable $y$ is SST or SIC, a is a constant, b is the regression coefficient, $\overline{x}$ is the average value of $x$, $\overline{y}$ is the average value of $y$, and $n$ is the number. The linear regression method was used to calculate the change trend of SST and SIC.

### 3.3. Correlation Analysis

We calculated the correlation between two variables to represent their degree of relation, the correlation coefficient with covariance between the variables, and the standard deviation of the business in the range of $(-1, 1)$, in which a negative correlation between them suggested a negative relation, a value of zero suggested no significant correlation between them, and a positive correlation suggested the two were positively related, thereby increasing or decreasing at the same time. The Pearson product moment correlation coefficient was used to represent the relationship between SST and SIC in the Artic, where $\overline{X}$ is the mean value of $X$ and $\overline{Y}$ is the mean value of $Y$.

$$r = \frac{\sum(X - \overline{X})(Y - \overline{Y})}{\left(\sqrt{\sum_{i=1}^{n}(X_i - \overline{X})^2}\right)\left(\sqrt{\sum_{i=1}^{n}(Y_i - \overline{Y})^2}\right)} \tag{5}$$

## 4. Definition of Sea Ice Melting and Freezing Processes

### 4.1. Sea Ice Break-Up and Melt Completely

The early and prolonged melting period of Arctic sea ice and ice snow has been one of the signals of accelerated Arctic climate warming in recent years [43]. SIC is an important indicator of sea ice melting and freezing [44]. Most studies define sea ice break-up as the date on which SIC falls below a specified threshold and remains below that threshold for a specified period of time [45]. We defined the sea ice break-up date as the first day when the SIC was less than 90% for five consecutive days, and the first day when the full melting state (open water) was less than 15% for five consecutive days [46]. The time period from the beginning of the melting state to the complete melting state is the rapid melting period. We used the summer months from April to August as the melting season [47].

### 4.2. Sea Ice Freeze-Up and Freeze Completely

Among the many problems faced by the opening of the Arctic route, the rapid change in Arctic sea ice has formed a huge challenge. In November 2021, in the Arctic waters near Russia, there were ships trapped in the ice and unable to sail, primarily because they had not anticipated the change in the freezing situation of the sea ice in the region and had to wait for rescue by the icebreaker. Sea ice freeze-up was defined as the first day on which the SIC was greater than 15% for five consecutive days, and the complete freezing state was defined as the first day on which the SIC was greater than 90% for five consecutive days [48]. The period from the start to the full freeze state is the fast freeze period. The freezing season will be from September this year to March next year.

## 5. Result and Analysis

### 5.1. Spatiotemporal Variation of SST and Sea Ice

There are significant spatial differences between Arctic SST and SIC. The annual variation of the Arctic SST shows that the overall warming trend is not obvious, but some sea areas are warming significantly, and the annual average SST is $-0.13\,°C$. The Norwegian Sea has the highest SST at $5.53\,°C$, followed by the Greenland Sea at $3.93\,°C$ and the Barents Sea at $2.16\,°C$. The average annual SST below $0\,°C$ is $-0.99\,°C$ in the Beaufort Sea, $-0.74\,°C$ in the Laptev Sea, $-0.73\,°C$ in the Northwest Passage, and $-0.56\,°C$ in the East Siberian Sea. Only the Kara Sea, Barents Sea, and Laptev Sea passed the significance test ($p < 0.05$), and the trends are $0.068\,°C/year$, $0.052\,°C/year$, and $0.044\,°C/year$, respectively. This indicates that the Northeast Passage is the area with the fastest rise in SST in the Arctic in recent years. The annual mean of Arctic SIC is approximately 50.29%, showing a significant downward trend of approximately $-0.31\%/year$. The concentration of sea ice in the Beaufort Sea was the highest (80.92%), followed by the Northwest Passage (72.47%), Laptev Sea (70.71%), East Siberian Sea (70.38%), Chukchi Sea (57.41%), and Kara Sea (54.73%). The concentration of sea ice was lower in Baffin Bay (38.51%), the Barents Sea (15.22%), the Greenland Sea (12.42%), and the Norwegian Sea (3.23%). Among them, the Kara Sea has the largest decreasing trend of SIC of approximately $-0.72\%/year$. The Laptev Sea, Chukchi Sea, Barents Sea, and Northwest Passage show a downward trend of $-0.64\%/year$, $-0.53\%/year$, $-0.41\%/year$, and $-0.26\%/year$, respectively. It can be seen that the Northeast Passage is the region with the fastest decline in SIC among the Arctic marginal waters in recent years.

The seasonal variation of SST is high in summer (June–August) and autumn (September–November) and low in winter (December–February) and spring (March–May).

The average temperature in winter, spring, summer, and autumn was $-0.79\,°C$, $-0.78\,°C$, $0.80\,°C$, and $0.35\,°C$, respectively. Only in autumn did the average temperature increase by $0.01\,°C/year$, and the other three seasons did not increase significantly. In winter, the Kara Sea had the most obvious warming trend (approximately $0.03\,°C/year\,°C$), followed by the Laptev Sea, East Siberian Sea, and Beaufort Sea. It is worth noting that, unlike other Arctic marginal seas, Baffin Bay has a cooling trend of $-0.02\,°C/year$ in winter. In spring, the Laptev Sea showed the largest warming trend (approximately $0.03\,°C/year$),

followed by the Kara Sea, East Siberian Sea, and Beaufort Sea. In summer, the temperature of all sea areas is above 0 °C (Figure 2). The Kara Sea has the highest Arctic warming trend of 0.11 °C/year, followed by the Barents Sea with 0.09 °C/year and the Laptev Sea with 0.05 °C/year. In autumn, the Kara Sea is the sea area with the largest warming trend of 0.11 °C/year, followed by the Barents Sea, Laptev Sea, and Norwegian Sea.

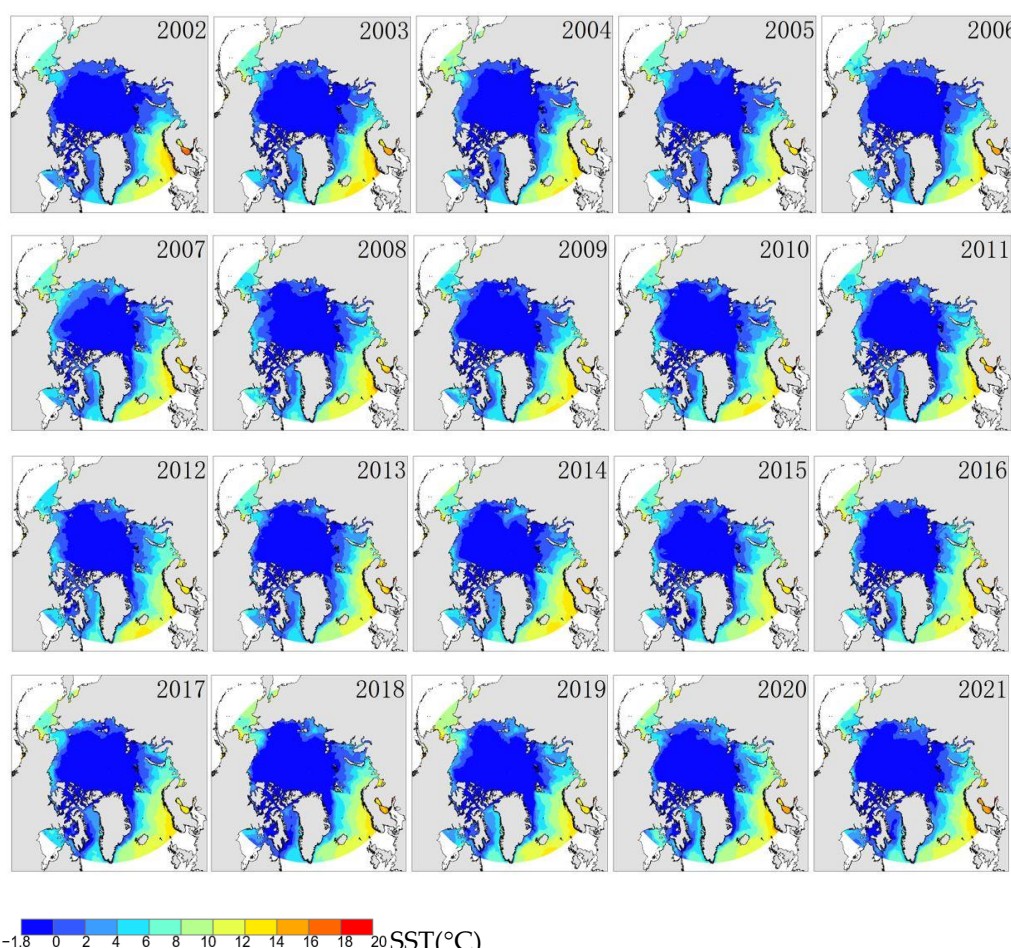

**Figure 2.** Arctic summer SST from June to August.

Arctic SIC is low in summer and autumn and high in winter and spring (Table 2). The concentration of sea ice in autumn is the lowest and the decreasing trend is the largest. The SIC in spring is the highest and the decreasing trend is the lowest. The trend of SIC in summer, autumn, winter, and spring is −0.39%/year, −0.45%/year, −0.24%/year, and −0.23%/year, respectively. In winter, the downward trend of SIC in the Kara Sea and Chukchi Sea reaches −0.55%/year, and the Norwegian Sea, Northwest Passage, and East Siberian Sea also show a significant downward trend of SIC. In spring, the SIC decreases by −0.81%/year in the Barents Sea, −0.59%/year in the Chukchi Sea, −0.44%/year in the Kara Sea, and −0.22%/year in the East Siberian Sea and Beaufort Sea. In summer and autumn, the sea ice in the Arctic marginal waters is the least, and the seasonal average SIC is less than 70%. The SIC in the Northeast Passage decreased significantly in summer and autumn. The SIC of the Barents Sea, Kara Sea, and Laptev Sea decreased by −0.33%/year, −1.32%/year, and −1.51%/year in summer and −0.38%/year, −1.38%/year, and −1.49%/year in autumn, respectively. In terms of seasonal changes, SIC in the Arctic is the smallest and decreases significantly in summer and autumn, and the decreasing trend is most significant in the Northeast Passage.

**Table 2.** Seasonal average SST (°C) and SIC (%) in typical sea areas of the Arctic, 2002–2021.

| Area | Winter | | Spring | | Summer | | Autumn | |
|---|---|---|---|---|---|---|---|---|
| | °C | % | °C | % | °C | % | °C | % |
| Norwegian Sea | 4.15 | 4.89 | 3.83 | 6.59 | 7.58 | 1.52 | 6.54 | 0.89 |
| Barents Sea | 0.49 | 25.89 | 0.18 | 30.14 | 4.40 | 4.58 | 3.56 | 3.77 |
| Kara Sea | −1.53 | 86.16 | −1.42 | 88.64 | 1.69 | 27.37 | 1.00 | 25.13 |
| Laptev Sea | −1.63 | 98.24 | −1.49 | 95.09 | 0.41 | 45.61 | −0.26 | 49.64 |
| East Siberian Sea | −1.48 | 91.23 | −1.37 | 90.29 | 0.52 | 55.44 | 0.08 | 48.41 |
| Chukchi Sea | −1.42 | 84.20 | −1.33 | 84.55 | 1.81 | 38.07 | 1.11 | 28.65 |
| Beaufort Sea | −1.64 | 98.66 | −1.51 | 95.83 | 0.01 | 64.42 | −0.82 | 68.64 |
| Northwest Passage | −1.60 | 97.01 | −1.47 | 94.00 | 0.39 | 53.77 | −0.23 | 50.01 |
| Baffin Bay | −0.81 | 65.80 | −0.93 | 69.56 | 2.48 | 16.52 | 1.50 | 10.37 |
| Greenland Sea | 2.78 | 17.01 | 2.63 | 17.35 | 5.56 | 7.44 | 4.74 | 9.23 |

On the monthly scale, the SST in the Arctic was low from November to June, and the SST in March was the lowest at approximately −0.94 °C (Figure 3). The SST is high from July to October, with the highest SST reaching approximately 1.55 °C in August, which is also the golden period for the passage of the Arctic shipping routes. In January, February, and May, the SST was above 0 °C in the Norwegian Sea, Barents Sea, and Greenland Sea, but all the other sea areas were below 0 °C. From March to April, only the Norwegian and Greenland Seas were above 0 °C. The Norwegian Sea, Barents Sea, Kara Sea, Greenland Sea, Baffin Bay, and Chukchi Sea showed the fastest warming in May-June, while the Laptev Sea showed the fastest warming in June-July. Since June, temperatures in the Chukchi Sea and Baffin Bay have risen above 0 °C. Although the SST of all the Arctic marginal seas is greater than 0 °C from July to September, the SST of all the Arctic seas shows a rapid decline from August to September, except for the Norwegian Sea and the Greenland Sea. From October to December, temperatures begin to cool again, but the Norwegian and Barents Seas remain above 0 °C. Arctic SIC is less than 50% from July to October and more than 50% from November to June. SIC was lowest in September at approximately 24% and highest in March at approximately 68% (Figure 4). February, April, June, October, October, May, September, July, June, and March show the greatest decline in SIC in the Norwegian Sea, Barents Sea, Kara Sea, Laptev Sea, East Siberian Sea, Chukchi Sea, Beaufort Sea, Northwest Passage, Baffin Bay, and Greenland Sea, respectively. Among them, the SIC of the Laptev Sea decreases the most, and the change trend of Arctic SIC has obvious regional characteristics.

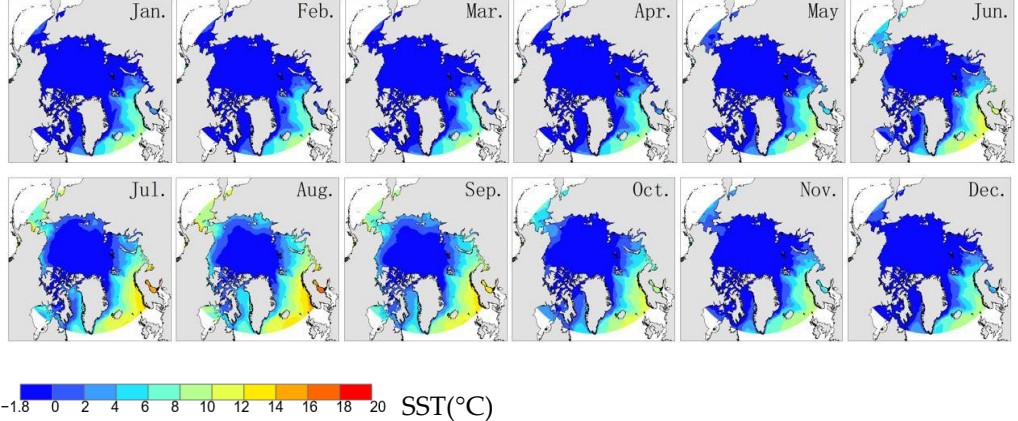

**Figure 3.** Monthly mean SST of Arctic from 2002 to 2021.

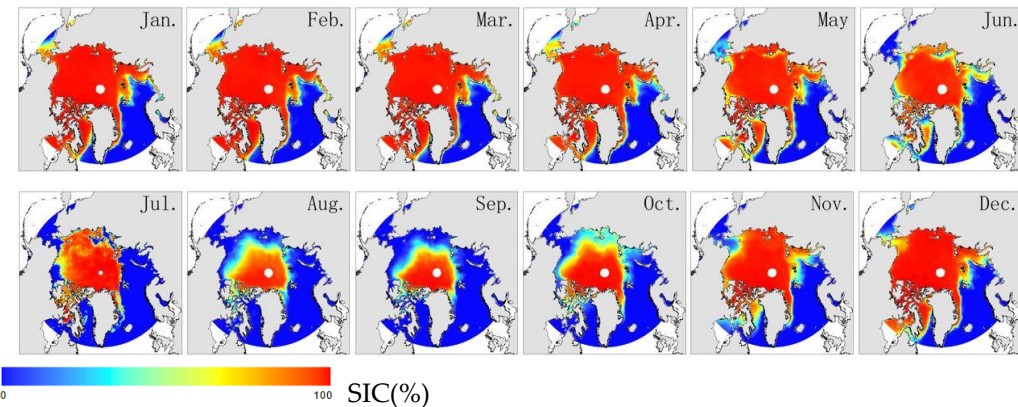

**Figure 4.** Monthly mean SIC in Arctic from 2002 to 2021.

*5.2. Response of SST to Sea Ice Concentration*

5.2.1. Correlation between SST and SIC

Within a pixel, there is both the SIC and SST, of which the SIC value represents the proportion of the area covered by sea ice and the SST represents the temperature of the remaining area. The same location can be used in different ways to indicate both SST and SIC. From 2002 to 2021, the annual average SST and SIC in the Arctic showed a significant trade-off (Table 3). From the waveform curve, the peaks of SST correspond to the troughs of SIC, the troughs of SST correspond to the peaks of SIC, and the intensity of the temperature changes is approximately the same as the change of SIC.

**Table 3.** SST and SIC in the Arctic, 2002–2021.

| Year | SST (°C) | SIC (%) |
|------|----------|---------|
| 2002 | −0.21 | 45.76 |
| 2003 | −0.16 | 55.30 |
| 2004 | −0.19 | 55.35 |
| 2005 | −0.09 | 53.49 |
| 2006 | −0.12 | 52.79 |
| 2007 | 0.01 | 50.16 |
| 2008 | −0.13 | 52.81 |
| 2009 | −0.20 | 53.12 |
| 2010 | −0.17 | 51.88 |
| 2011 | −0.13 | 47.89 |
| 2012 | 0.01 | 36.71 |
| 2013 | −0.06 | 52.59 |
| 2014 | −0.05 | 51.85 |
| 2015 | −0.05 | 50.99 |
| 2016 | 0.03 | 47.97 |
| 2017 | −0.12 | 49.84 |
| 2018 | −0.16 | 49.71 |
| 2019 | −0.12 | 48.55 |
| 2020 | 0.15 | 48.11 |
| 2021 | −0.22 | 50.91 |

In addition, the Northeast Passage and Northwest Passage are vital for Arctic shipping. The ice conditions in the passage areas are complicated, and the changes in sea surface temperature and sea ice density are also sensitive. Four points were selected in the Kara Sea (68°0′49.07″N, 45°8′58.54″E), the Laptev Sea (79°4′58.08″N, 79°7′17.00″E), the Eastern Siberian Sea (74°1′51.64″N, 142°0′56.00″E), and the Northwest Passage (74°2′13.40″N, 97°5′58.00″W). One can observe how SST and SIC changed at each point in 2021 (Figure 5). From March to May before the summer melt season, the SIC changes repeatedly, and the sea ice underwent a freezing–melting–refreezing process until the real summer came when

the sea ice completely melted, and by November, the sea ice froze again. This freeze–thaw phenomenon occurs in May–July when the sea ice begins to freeze again into winter in September after approximately two months of thawing. The sample in the East Siberian Sea is approximately the same as the sample in the Laptev Sea, but the freeze–thaw process is earlier and lasts for a shorter time around May–June. The Northwest Passage sample site is completely different from the other three sites, and the freeze–thaw process of sea ice occurs in March–May and the ice-free period is earlier and of a shorter duration. Starting in mid-June, over the following four months, the freeze–melt–refreeze process continued until the sea ice completely froze in mid-October. We found that there was a period of freeze–thaw–thaw cycles before and for some time after the complete melting of summer sea ice. During the sea ice change process before the complete melting of sea ice, the SST increases, and during the sea ice change process after the complete melting of sea ice, the SST decreases. Of course, these phenomena are related to the latitude of the four selected points and the position of the sea and land. To further explore this difference across Arctic seas and the relationship between SST and changes in SIC, we calculated correlation coefficients in the Arctic.

SST has a significant negative correlation with SIC (Figure 1), and the overall correlation coefficient of the study area is −0.82, which shows obvious regional differences under the influence of latitude and warm current (Figure 6). However, it is worth noting that the SST data are uniformly expressed as −1.8 °C at the pixel points where they are completely covered by ice. This may lead to a high correlation between SST and SIC. The negative correlation between SST and SIC is related to the polar amplification phenomenon. The loss of sea ice exposes the ocean to sunlight, and the ocean has a lower albedo, which absorbs more solar radiation and further promotes the melting of sea ice, forming positive feedback [49].

The correlation coefficients of the Norwegian Sea, Greenland Sea, and Barents Sea were −0.42, −0.59, and −0.67, respectively, showing a low correlation (Figure 7). On the side close to the North Atlantic, the correlation between SST and SIC is small, and even fails to pass the significance test of correlation, showing no correlation. The heat from the Atlantic enters the Arctic Ocean primarily through the Barents Sea, the Fram Strait, and the Davis Strait [50–52]. Since this part of the ocean is a conduit for warm water from the North Atlantic to the Arctic Ocean, eastern Greenland blocks the direct exchange of warm water with the North Pole, allowing warmer water to flow into this part of the ocean and bring more heat with it. As a result, SST has been high, there is relatively less sea ice, and the direct effect of SST on sea ice loss is also reduced. Other factors, such as ocean currents, air pressure, wind speed, and precipitation, are likely to play a stronger role in sea ice changes. The correlation coefficient between SST and SIC is −0.80 in Baffin Bay, which is located the west of the Greenland Sea. The southern part of Baffin Bay is more affected by the North Atlantic Warm current and the correlation coefficient is lower. The northern part is relatively more closed and less affected. The correlation coefficients of the Chukchi Sea, Beaufort Sea, and Northwest Passage are −0.91, −0.92, and −0.90, respectively. This part of the sea area is relatively closed, and the correlation between SST and SIC is enhanced. The Chukchi Sea is influenced by the warm waters of the Pacific Ocean flowing in through the Bering Strait [53], with less correlation in the south than in the north. The Beaufort Sea is located between the Chukchi Sea and the Northwest Passage, and the energy exchange between SST and sea ice is less affected by other factors. The waters of the Northwest Passage are characterized by small passages, and the Baffin Bay to the east cushions the influence of the North Atlantic Current. The correlation coefficients between SIC and SST in the Kara Sea, Laptev Sea, and East Siberian Sea are −0.93, −0.96, and −0.96, respectively, which are the regions with the largest correlation coefficients between SST and SIC in the Arctic marginal seas. Compared with other Arctic marginal seas, these sea areas are more closed. It is also the key sea area that determines whether the Northeast Passage is smoothly navigable. The negative correlation between SST and SIC is approximately −0.80 in the Kara Sea area north of Novaya Scala and near the Kara Strait because the warm water of

the North Atlantic Ocean flows into the Kara Sea from the Kara Strait and north of Novaya Scala through the Barents Sea, which may be the reason for the relatively weak negative correlation in this part of the sea area. The Laptev Sea is a relatively closed sea area in the Northeast Passage, and there is a significant negative correlation between SST and SIC. The negative correlation between SST and SIC in the East Siberian Sea is strong in the east but weak in the west. This is due to the warm waters of the Pacific Ocean, which flow through the Chukchi Sea and enter from the eastern part of the East Siberian Sea, causing it to be affected.

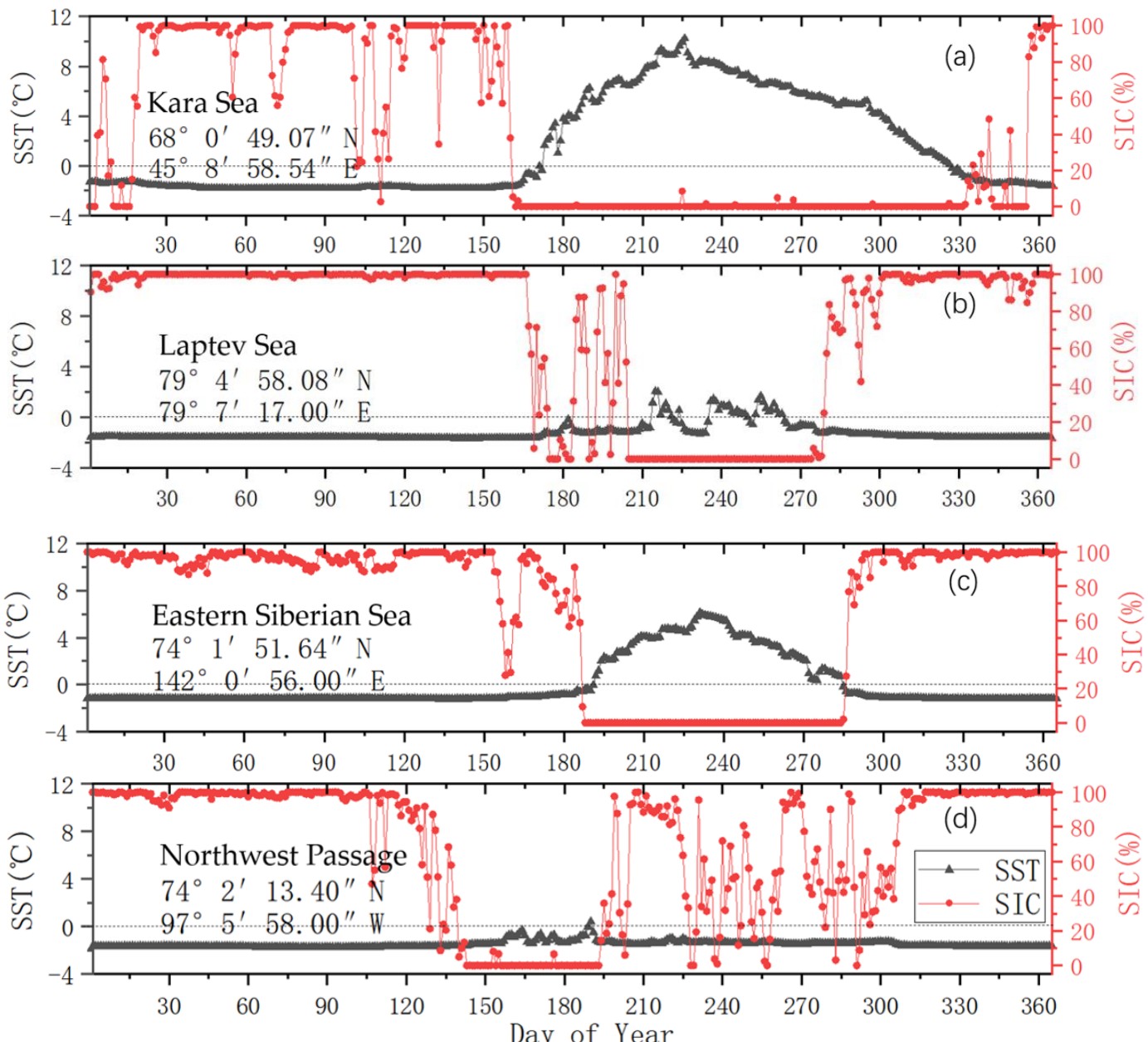

**Figure 5.** SST and SIC in the Arctic in 2021. The black line connected by the triangle is SST and the red line connected by the circle is sea ice concentration. (**a–d**) are points in the Kara Sea, Laptev Sea, East Siberian Sea, and Northwest Passage, respectively, and their corresponding coordinates are shown in the figure.

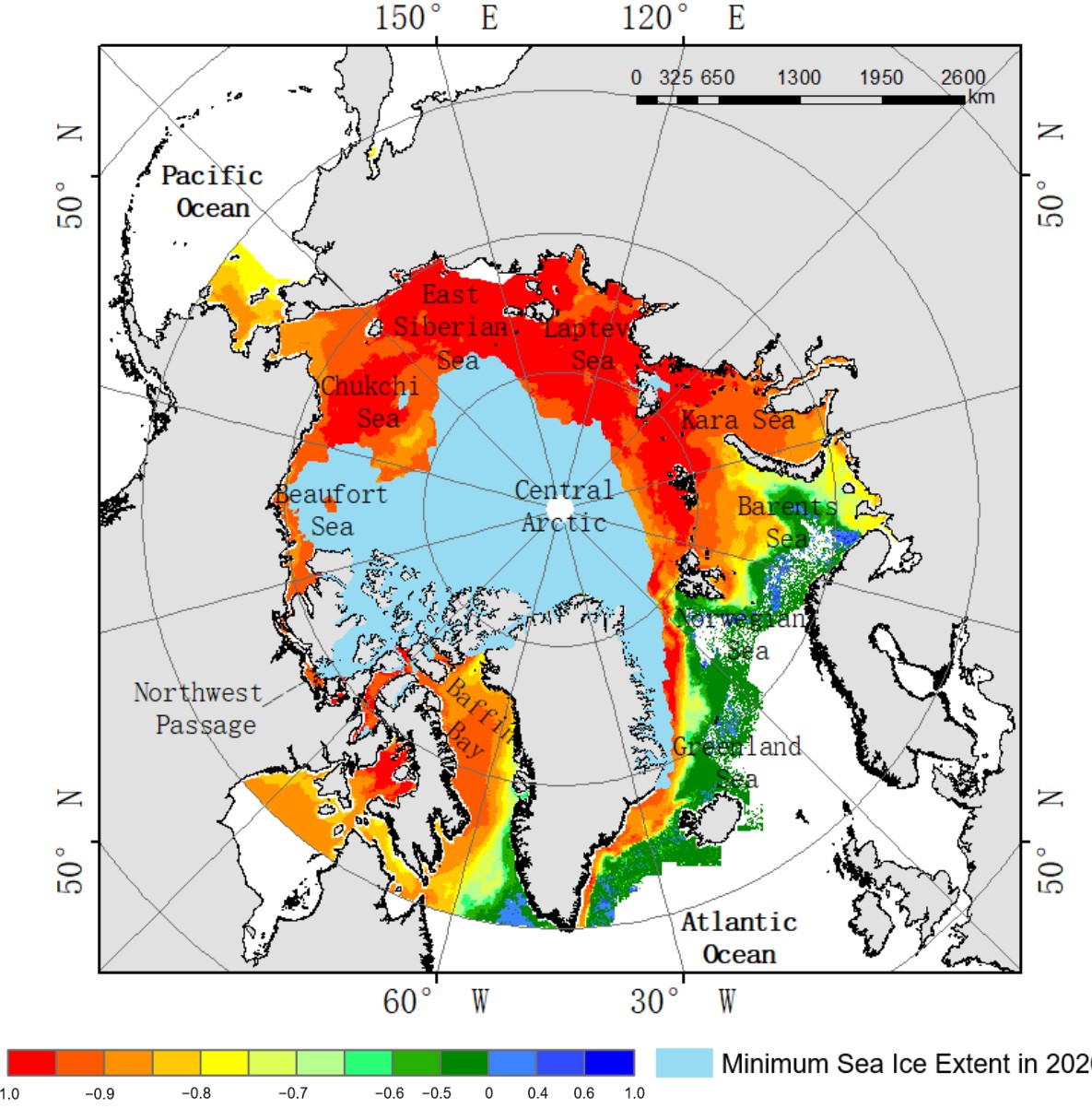

**Figure 6.** Correlation coefficient of SIC and SST in the Arctic from 2002 to 2021.

Sea ice loss interacts with an increase in SST, especially in summer and fall. The rise in SST may delay the ice sealing in autumn, resulting in thinner sea ice, which is more likely to melt in the following summer, and the formation of an open water surface further promotes the rise in SST. As temperatures rise and sea ice melts, summer SIC values fall. This indicates that SST is closely related to sea ice and can be used as an effective parameter for sea ice prediction.

### 5.2.2. Melting and Freezing Processes of Sea Ice

In recent years, Arctic sea ice has continued to decrease and glacier melting has accelerated, resulting in the situation that Arctic sea ice melts earlier in summer and freezes later in autumn [54]. The melting and freezing of sea ice play a regulating role in the balance of surface material and energy, affecting the global water cycle process and the change in sea level, reflecting the changes in the Arctic environment, and affecting the arctic resources for mining and transportation, marine mammal migration, the arctic coastal countries' residents living on the ice, and arctic shipping safety. Therefore, we further studied the thawing period of the Arctic marginal waters.

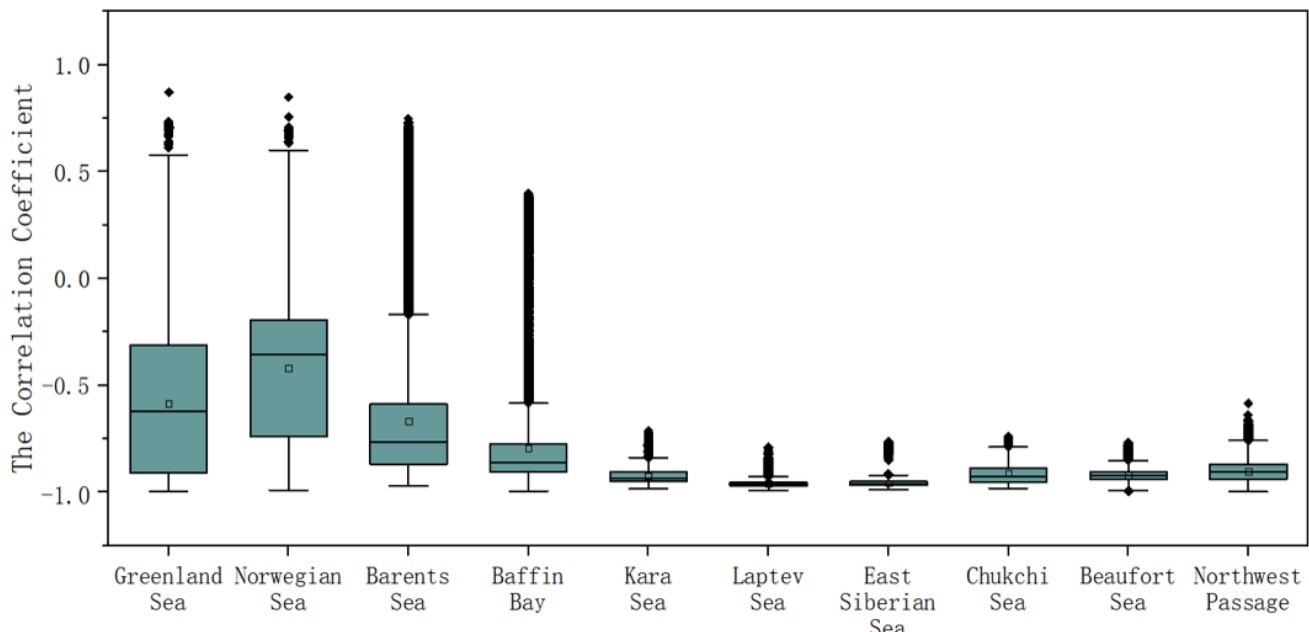

**Figure 7.** Correlation coefficient of SIC and SST in the Arctic marginal sea areas from 2002 to 2021.

The Arctic sea ice break-up occurs from early April to the end of August each year (Figure 8), from approximately DoY 91 to 239 (Table 4). The break-up occurs, on average, in mid to late May on DoY 143 or May 23. The time to complete melting is around early June (Figure 9) on DoY 159 or June 8 (Table 5). The duration of melting varies greatly by sea area, with the Norwegian Sea beginning to melt the earliest on around April 6 (DoY 96). The Barents Sea and Greenland Sea ice break-up occurs on DoYs 105 and 102, respectively. This was followed by Baffin Bay and the Kara Sea, which started melting on DoYs 135 and 149. The Northwest Passage break-up occurs at similar times as the Laptev Sea on DoYs 162 and 166, respectively. The Beaufort Sea and the East Siberian Sea ice break-up occurs last, on or around June 25 (DoY 176). The Norwegian Sea reached full melting the earliest, on average on DoY 98, and was the fastest in the Arctic. The Greenland Sea and the Barents Sea melted on DoYs 108 and 115, respectively. Baffin Bay reached full melting on DoY 156. The Chukchi Sea and the Kara Sea were close on DoYs 170 and 177, respectively. The Beaufort Sea, the Northwest Passage, the East Siberian Sea, and the Laptev Sea were the last to reach full melting on DoYs 188, 191, 191, and 193, respectively.

**Table 4.** DoY of Arctic sea ice break-up from 2002 to 2020.

| Year | Norwegian Sea | Barents Sea | Kara Sea | Laptev Sea | East Siberian Sea | Chukchi Sea | Beaufort Sea | Northwest Passage | Baffin Bay | Greenland Sea |
|------|------|------|------|------|------|------|------|------|------|------|
| 2002 | 96  | 119 | 167 | 171 | 179 | 161 | 184 | 167 | 132 | 101 |
| 2003 | 100 | 118 | 162 | 188 | 183 | 164 | 179 | 171 | 140 | 105 |
| 2004 | 96  | 113 | 162 | 170 | 180 | 167 | 177 | 163 | 136 | 103 |
| 2005 | 94  | 99  | 156 | 178 | 173 | 168 | 199 | 163 | 133 | 102 |
| 2006 | 97  | 105 | 147 | 152 | 169 | 149 | 158 | 162 | 137 | 103 |
| 2007 | 95  | 103 | 147 | 178 | 186 | 164 | 157 | 167 | 138 | 101 |
| 2008 | 97  | 107 | 158 | 172 | 185 | 171 | 181 | 172 | 138 | 102 |
| 2009 | 98  | 110 | 149 | 170 | 187 | 171 | 172 | 157 | 130 | 105 |
| 2010 | 97  | 106 | 139 | 163 | 181 | 161 | 173 | 163 | 133 | 103 |
| 2012 | 98  | 106 | 148 | 162 | 177 | 165 | 185 | 156 | 133 | 106 |
| 2013 | 96  | 105 | 158 | 154 | 185 | 161 | 176 | 168 | 134 | 103 |
| 2014 | 96  | 102 | 139 | 176 | 177 | 156 | 174 | 153 | 144 | 102 |
| 2015 | 94  | 96  | 137 | 168 | 178 | 159 | 156 | 157 | 136 | 102 |

**Table 4.** *Cont.*

| Year | Norwegian Sea | Barents Sea | Kara Sea | Laptev Sea | East Siberian Sea | Chukchi Sea | Beaufort Sea | Northwest Passage | Baffin Bay | Greenland Sea |
|------|---------------|-------------|----------|------------|-------------------|-------------|--------------|-------------------|------------|---------------|
| 2016 | 94 | 101 | 136 | 158 | 171 | 156 | 179 | 161 | 137 | 104 |
| 2017 | 93 | 106 | 162 | 160 | 180 | 154 | 179 | 164 | 138 | 100 |
| 2018 | 95 | 102 | 137 | 158 | 171 | 150 | 165 | 162 | 131 | 102 |
| 2019 | 94 | 102 | 136 | 148 | 161 | 159 | 168 | 152 | 125 | 99 |
| 2020 | 93 | 97 | 142 | 157 | 166 | 169 | 184 | 159 | 128 | 98 |
| 2021 | 100 | 119 | 167 | 171 | 179 | 161 | 184 | 167 | 132 | 101 |

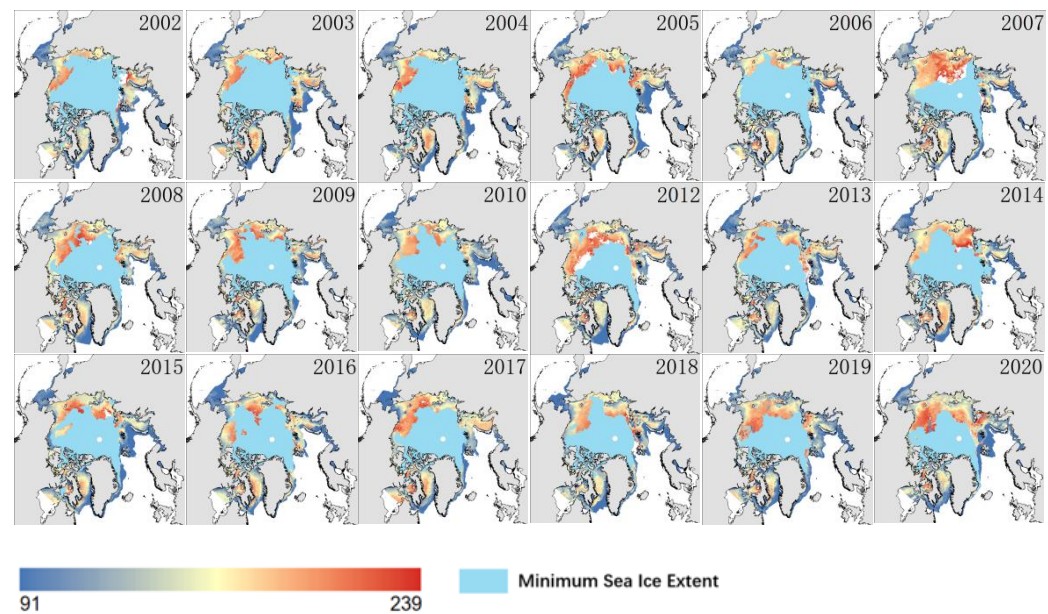

**Figure 8.** Spatial distribution of Arctic sea ice break-up dates (DoY) from 2002 to 2020.

**Table 5.** DoY of Arctic sea ice complete melt from 2002 to 2020.

| Year | Norwegian Sea | Barents Sea | Kara Sea | Laptev Sea | East Siberian Sea | Chukchi Sea | Beaufort Sea | Northwest Passage | Baffin Bay | Greenland Sea |
|------|---------------|-------------|----------|------------|-------------------|-------------|--------------|-------------------|------------|---------------|
| 2002 | 101 | 136 | 198 | 193 | 193 | 161 | 189 | 186 | 154 | 111 |
| 2003 | 102 | 132 | 186 | 215 | 198 | 174 | 192 | 194 | 161 | 109 |
| 2004 | 98 | 125 | 183 | 199 | 192 | 166 | 174 | 193 | 158 | 107 |
| 2005 | 96 | 107 | 179 | 201 | 195 | 169 | 205 | 184 | 152 | 107 |
| 2006 | 99 | 114 | 176 | 185 | 187 | 175 | 190 | 195 | 157 | 109 |
| 2007 | 98 | 112 | 181 | 206 | 204 | 181 | 177 | 196 | 158 | 109 |
| 2008 | 100 | 119 | 187 | 200 | 196 | 179 | 196 | 193 | 158 | 105 |
| 2009 | 99 | 121 | 179 | 190 | 189 | 181 | 184 | 190 | 152 | 111 |
| 2010 | 100 | 117 | 166 | 186 | 186 | 173 | 188 | 194 | 157 | 104 |
| 2012 | 100 | 114 | 178 | 191 | 190 | 173 | 201 | 196 | 157 | 112 |
| 2013 | 99 | 115 | 188 | 184 | 187 | 170 | 184 | 190 | 157 | 110 |
| 2014 | 98 | 108 | 163 | 199 | 187 | 168 | 194 | 193 | 165 | 108 |
| 2015 | 96 | 103 | 159 | 202 | 197 | 172 | 176 | 188 | 153 | 107 |
| 2016 | 97 | 112 | 173 | 196 | 177 | 161 | 189 | 186 | 162 | 112 |
| 2017 | 95 | 116 | 183 | 189 | 189 | 157 | 194 | 190 | 161 | 104 |
| 2018 | 98 | 110 | 172 | 186 | 194 | 155 | 174 | 192 | 149 | 109 |
| 2019 | 98 | 112 | 158 | 177 | 181 | 177 | 196 | 187 | 148 | 106 |
| 2020 | 94 | 105 | 169 | 179 | 188 | 166 | 183 | 188 | 155 | 107 |
| 2021 | 101 | 136 | 198 | 193 | 193 | 161 | 189 | 186 | 154 | 111 |

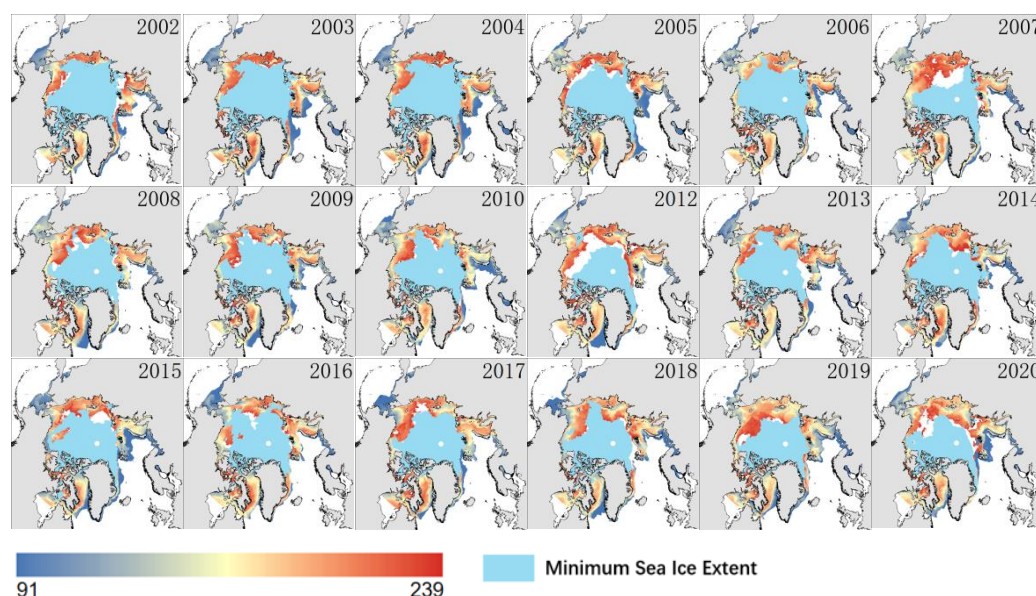

**Figure 9.** Spatial distribution of Arctic sea ice complete melt dates (DoY) from 2002 to 2020.

The Arctic freezes from early September to the end of December (Figure 10), around DoY 245–365 (Table 6), and the onset of the freeze, on average, occurs in late October, around DoY 296 or October 23. The time to complete freeze is in early November, on DoY 305 or around 1 November. In the Arctic, the Beaufort Sea freeze-up occurs first on September 23 (DoY 266) and reaches full freeze the fastest on DoY 277. The Laptev Sea, East Siberian Sea, Northwest Passage, and Greenland Sea freeze-up and complete freeze occur at the same time (Table 7), starting to freeze on DoYs 282, 283, 283, and 287, respectively, and completely freeze on DoYs 292, 291, 295, and 297, respectively (Figure 11). The Kara Sea and Chukchi Sea freeze-up occurs on DoYs 305 and 303, respectively, and reaches full freeze on DoYs 315 and 309. In the Norwegian Sea and Baffin Bay, on either side of Greenland, freeze-up occurs on DoYs 313 and 319, respectively, and they reach full freeze on DoYs 317 and 327. The Barents Sea was the last to begin freezing, on November 19 (DoY 323), and the last to reach full freeze, on DoY 329.

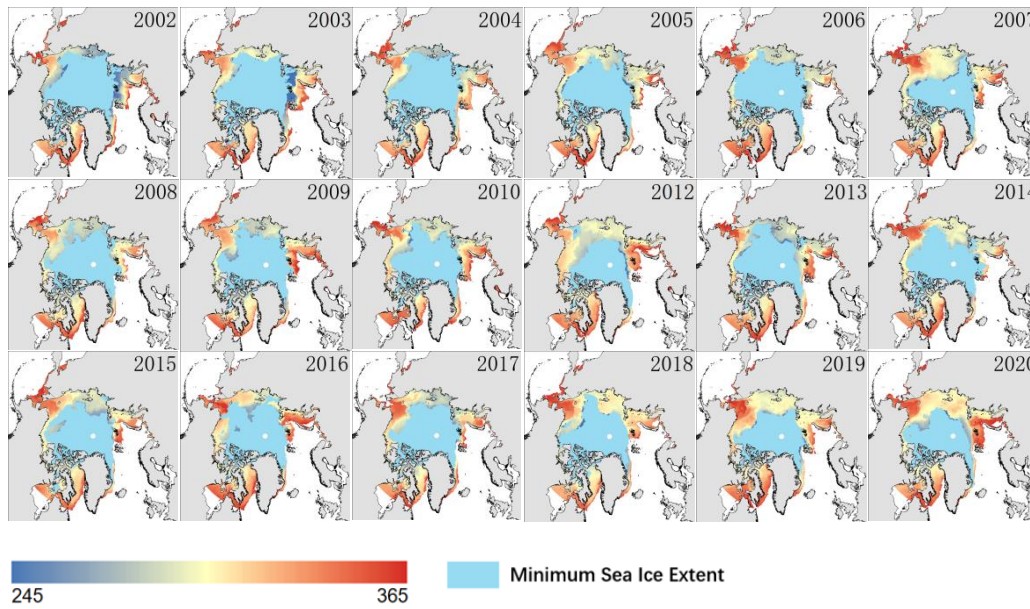

**Figure 10.** Spatial distribution of Arctic sea ice freeze-up dates (DoY) from 2002 to 2020.

**Table 6.** The DoY of Arctic sea ice freeze-up from 2002 to 2020.

| Year | Norwegian Sea | Barents Sea | Kara Sea | Laptev Sea | East Siberian Sea | Chukchi Sea | Beaufort Sea | Northwest Passage | Baffin Bay | Greenland Sea |
|------|-----------|---------|------|--------|---------------|---------|----------|---------------|----------|-----------|
| 2002 | 311 | 304 | 276 | 261 | 268 | 286 | 256 | 274 | 316 | 297 |
| 2003 | 313 | 308 | 280 | 270 | 277 | 291 | 258 | 276 | 314 | 295 |
| 2004 | 323 | 314 | 293 | 258 | 270 | 300 | 259 | 273 | 317 | 294 |
| 2005 | 305 | 315 | 302 | 284 | 283 | 297 | 256 | 278 | 317 | 279 |
| 2006 | 307 | 305 | 291 | 274 | 275 | 296 | 261 | 290 | 327 | 279 |
| 2007 | 315 | 325 | 308 | 280 | 303 | 316 | 264 | 285 | 316 | 272 |
| 2008 | 303 | 313 | 305 | 272 | 284 | 305 | 269 | 284 | 317 | 290 |
| 2009 | 301 | 337 | 315 | 282 | 275 | 303 | 260 | 282 | 313 | 277 |
| 2010 | 326 | 329 | 309 | 283 | 273 | 304 | 270 | 290 | 320 | 294 |
| 2012 | 318 | 334 | 322 | 293 | 292 | 304 | 287 | 292 | 322 | 274 |
| 2013 | 333 | 333 | 303 | 280 | 264 | 293 | 259 | 280 | 318 | 292 |
| 2014 | 299 | 302 | 296 | 292 | 280 | 300 | 264 | 278 | 322 | 286 |
| 2015 | 302 | 335 | 307 | 286 | 287 | 303 | 272 | 284 | 313 | 279 |
| 2016 | 290 | 336 | 331 | 283 | 293 | 307 | 281 | 289 | 325 | 283 |
| 2017 | 329 | 323 | 307 | 275 | 283 | 309 | 275 | 281 | 313 | 303 |
| 2018 | 314 | 331 | 312 | 301 | 282 | 312 | 260 | 276 | 311 | 289 |
| 2019 | 304 | 322 | 307 | 293 | 296 | 319 | 275 | 288 | 328 | 289 |
| 2020 | 336 | 343 | 327 | 306 | 302 | 313 | 263 | 287 | 324 | 286 |
| 2021 | 311 | 304 | 276 | 261 | 268 | 286 | 256 | 274 | 316 | 297 |

**Table 7.** The DoY of Arctic sea ice complete freeze from 2002 to 2020.

| Year | Norwegian Sea | Barents Sea | Kara Sea | Laptev Sea | East Siberian Sea | Chukchi Sea | Beaufort Sea | Northwest Passage | Baffin Bay | Greenland Sea |
|------|-----------|---------|------|--------|---------------|---------|----------|---------------|----------|-----------|
| 2002 | 311 | 308 | 287 | 271 | 272 | 293 | 264 | 286 | 325 | 303 |
| 2003 | 314 | 312 | 289 | 282 | 287 | 299 | 272 | 289 | 319 | 300 |
| 2004 | 328 | 317 | 304 | 267 | 277 | 305 | 266 | 283 | 325 | 302 |
| 2005 | 309 | 316 | 310 | 292 | 289 | 304 | 264 | 290 | 325 | 287 |
| 2006 | 292 | 310 | 301 | 285 | 284 | 302 | 275 | 302 | 332 | 290 |
| 2007 | 311 | 331 | 316 | 291 | 309 | 323 | 278 | 296 | 323 | 282 |
| 2008 | 308 | 318 | 315 | 283 | 290 | 311 | 282 | 297 | 327 | 300 |
| 2009 | 310 | 346 | 327 | 293 | 284 | 309 | 270 | 294 | 324 | 283 |
| 2010 | 336 | 336 | 321 | 294 | 289 | 309 | 281 | 301 | 330 | 302 |
| 2012 | 316 | 340 | 329 | 300 | 297 | 311 | 295 | 302 | 329 | 290 |
| 2013 | 348 | 341 | 316 | 291 | 268 | 296 | 266 | 292 | 328 | 306 |
| 2014 | 307 | 311 | 309 | 300 | 293 | 306 | 276 | 292 | 331 | 297 |
| 2015 | 289 | 342 | 320 | 294 | 293 | 306 | 282 | 297 | 327 | 294 |
| 2016 | 286 | 345 | 340 | 298 | 302 | 313 | 292 | 302 | 334 | 295 |
| 2017 | 342 | 332 | 317 | 284 | 291 | 314 | 285 | 292 | 325 | 312 |
| 2018 | 341 | 334 | 322 | 308 | 297 | 320 | 271 | 289 | 321 | 301 |
| 2019 | 315 | 331 | 318 | 301 | 301 | 326 | 284 | 299 | 334 | 301 |
| 2020 | 349 | 350 | 335 | 315 | 309 | 319 | 275 | 298 | 331 | 298 |
| 2021 | 311 | 308 | 287 | 271 | 272 | 293 | 264 | 286 | 325 | 303 |

The rapid melting (Figures 12 and 13) and freezing (Figures 14 and 15) processes lasted approximately 27 (Table 8) and 14 days (Table 9), respectively. However, there is no significant change in the rapid melting and freezing process. The Greenland Sea had the longest rapid melting period with 37 days, followed by the Northwest Passage with 36 days. The Laptev Sea, Kara Sea, East Siberian Sea, Beaufort Sea, Norwegian Sea, and Barents Sea experienced 31, 29, 26, 25, 23, and 22 days, respectively. The Chukchi Sea was the shortest, needing only 18 days to reach a fully melted state from the start. It takes only 11 days for the Laptev Sea and East Siberian Sea to reach full freeze, approximately 12 days for the Kara Sea and Chukchi Sea, 13 days for the Barents Sea, 14 days for Baffin Bay and

the Beaufort Sea, 15 days for the Northwest Passage, and 18 days for the Greenland Sea to rapidly freeze. In the Norwegian Sea, it takes 20 days.

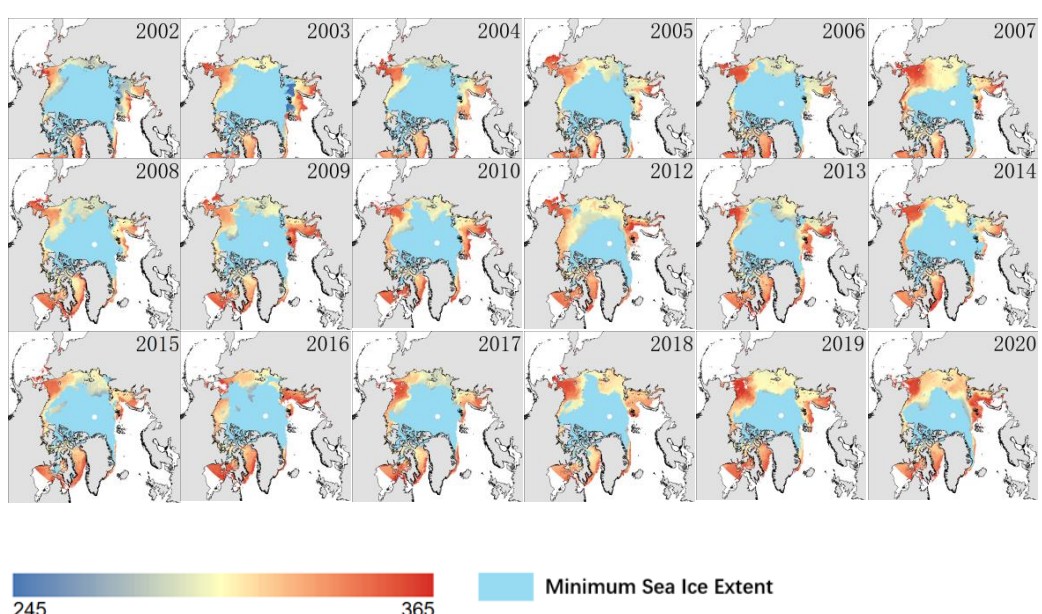

**Figure 11.** Spatial distribution of Arctic sea ice complete freeze dates (DoY) from 2002 to 2020.

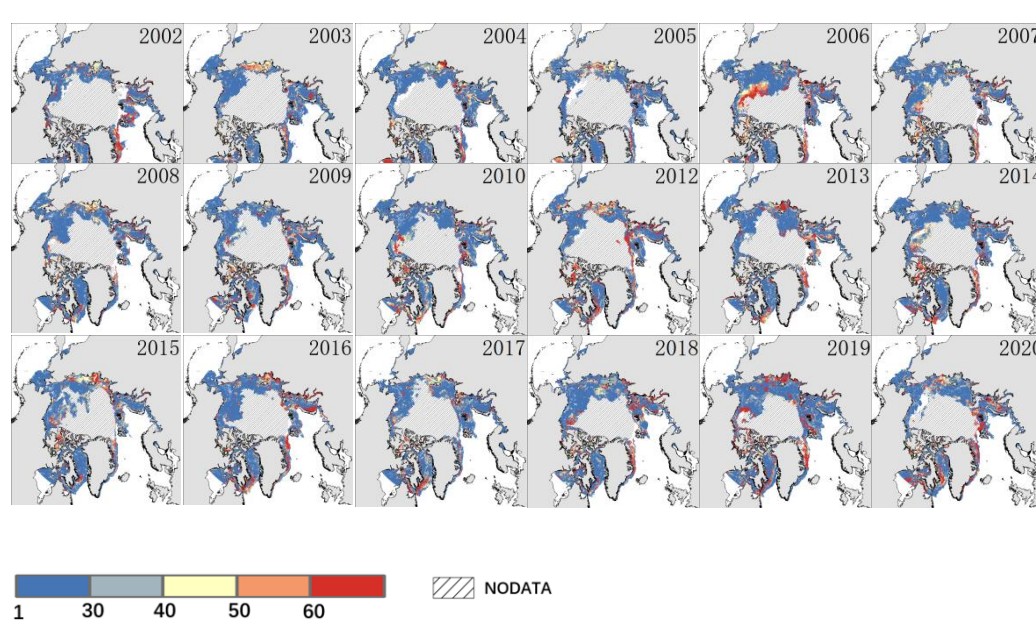

**Figure 12.** Number of days from breaking up to complete melting of Arctic sea ice, 2002–2020.

**Table 8.** Days from breaking up to complete melting of Arctic sea ice, 2002–2020.

| Year | Norwegian Sea | Barents Sea | Kara Sea | Laptev Sea | East Siberian Sea | Chukchi Sea | Beaufort Sea | Northwest Passage | Baffin Bay | Greenland Sea |
|------|---------------|-------------|----------|------------|-------------------|-------------|--------------|-------------------|------------|---------------|
| 2002 | 30 | 25 | 33 | 28 | 26 | 18 | 26 | 31 | 28 | 45 |
| 2003 | 20 | 22 | 24 | 37 | 29 | 13 | 22 | 33 | 28 | 25 |
| 2004 | 18 | 24 | 22 | 32 | 23 | 16 | 16 | 42 | 29 | 36 |
| 2005 | 12 | 15 | 23 | 27 | 35 | 19 | 19 | 30 | 25 | 29 |
| 2006 | 23 | 23 | 32 | 33 | 19 | 29 | 36 | 37 | 26 | 45 |
| 2007 | 28 | 26 | 34 | 35 | 24 | 20 | 28 | 35 | 23 | 44 |

**Table 8.** *Cont.*

| Year | Norwegian Sea | Barents Sea | Kara Sea | Laptev Sea | East Siberian Sea | Chukchi Sea | Beaufort Sea | Northwest Passage | Baffin Bay | Greenland Sea |
|------|------|------|------|------|------|------|------|------|------|------|
| 2008 | 26 | 24 | 30 | 31 | 23 | 15 | 25 | 30 | 23 | 29 |
| 2009 | 13 | 17 | 30 | 25 | 22 | 17 | 23 | 37 | 29 | 35 |
| 2010 | 23 | 20 | 28 | 24 | 23 | 20 | 34 | 37 | 31 | 34 |
| 2012 | 19 | 18 | 31 | 33 | 34 | 19 | 24 | 44 | 32 | 31 |
| 2013 | 30 | 26 | 32 | 31 | 24 | 18 | 17 | 30 | 27 | 41 |
| 2014 | 22 | 19 | 25 | 24 | 19 | 20 | 31 | 43 | 25 | 46 |
| 2015 | 21 | 23 | 25 | 42 | 25 | 19 | 26 | 34 | 21 | 35 |
| 2016 | 27 | 30 | 38 | 42 | 20 | 14 | 14 | 31 | 29 | 38 |
| 2017 | 20 | 20 | 22 | 30 | 30 | 17 | 26 | 32 | 29 | 27 |
| 2018 | 30 | 26 | 37 | 31 | 28 | 15 | 27 | 39 | 23 | 46 |
| 2019 | 32 | 20 | 23 | 29 | 28 | 22 | 43 | 41 | 29 | 48 |
| 2020 | 21 | 20 | 30 | 23 | 30 | 17 | 19 | 36 | 33 | 39 |
| 2021 | 30 | 25 | 33 | 28 | 26 | 18 | 26 | 31 | 28 | 45 |

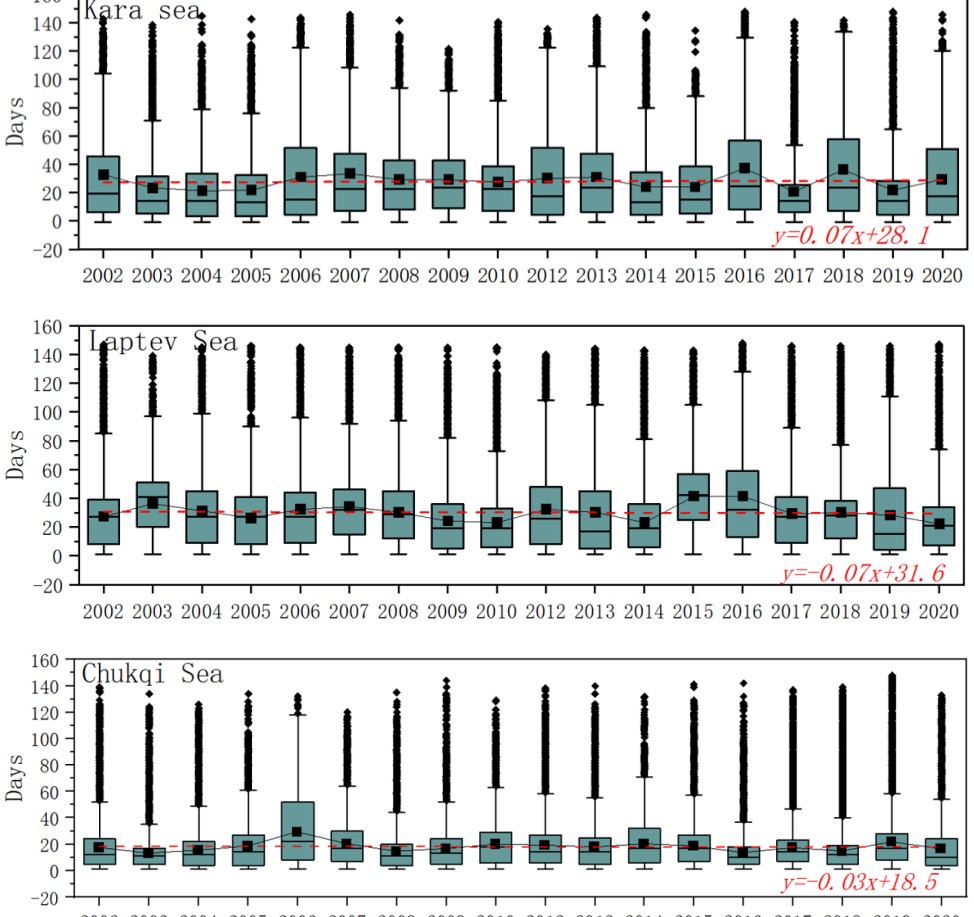

**Figure 13.** *Cont.*

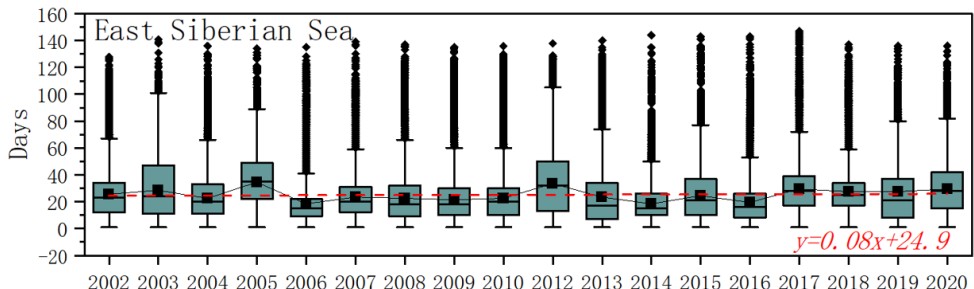

**Figure 13.** Number of days from breaking up to complete melting of sea ice in Kara Sea, Laptev Sea, East Siberian Sea, and Chukchi Sea from 2002 to 2020. The red dotted line is a linear fit of the mean days. The standard errors were 0.24, 0.26, 0.21, and 0.17, respectively, and passed the significance test ($p < 0.05$).

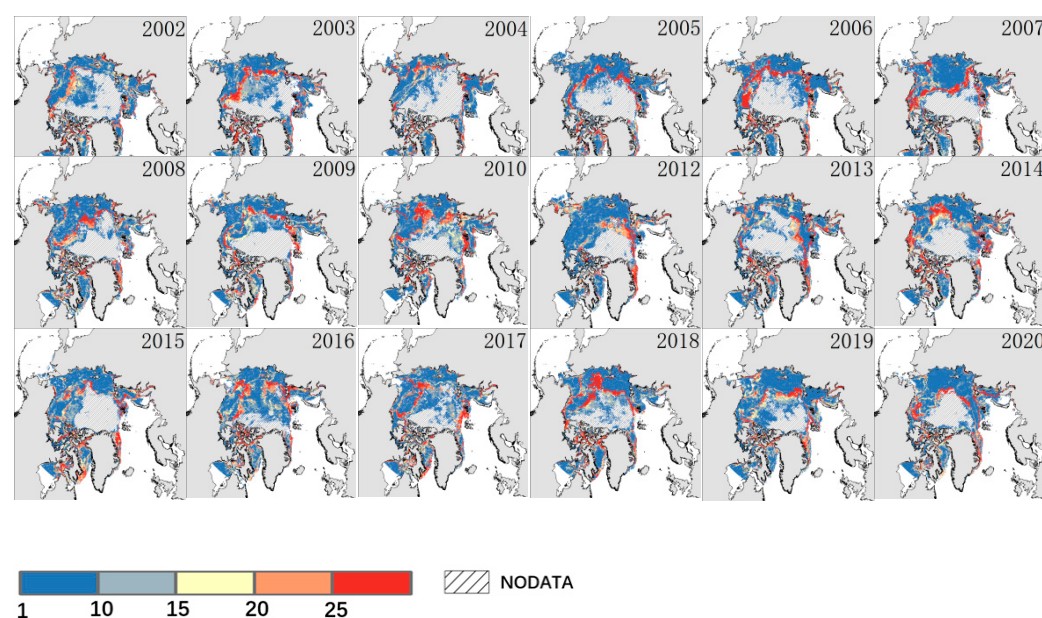

**Figure 14.** Number of days from freezing up to complete freezing of Arctic sea ice, 2002–2020.

**Table 9.** Days from freezing up to complete freezing of Arctic sea ice, 2002–2020.

| Year | Norwegian Sea | Barents Sea | Kara Sea | Laptev Sea | East Siberian Sea | Chukchi Sea | Beaufort Sea | Northwest Passage | Baffin Bay | Greenland Sea |
|------|------|------|------|------|------|------|------|------|------|------|
| 2002 | 14 | 13 | 12 | 12 | 9 | 12 | 11 | 15 | 14 | 14 |
| 2003 | 13 | 12 | 12 | 12 | 12 | 11 | 18 | 18 | 12 | 14 |
| 2004 | 9 | 9 | 11 | 14 | 12 | 10 | 11 | 15 | 12 | 16 |
| 2005 | 16 | 11 | 10 | 9 | 11 | 11 | 14 | 15 | 14 | 16 |
| 2006 | 12 | 13 | 10 | 11 | 13 | 16 | 21 | 16 | 12 | 21 |
| 2007 | 15 | 15 | 11 | 12 | 7 | 12 | 21 | 13 | 13 | 19 |
| 2008 | 21 | 14 | 13 | 13 | 9 | 11 | 14 | 15 | 14 | 19 |
| 2009 | 35 | 16 | 12 | 12 | 12 | 9 | 14 | 15 | 16 | 16 |
| 2010 | 20 | 15 | 12 | 11 | 17 | 10 | 13 | 16 | 15 | 16 |
| 2012 | 18 | 12 | 15 | 8 | 7 | 9 | 8 | 12 | 12 | 24 |
| 2013 | 16 | 12 | 14 | 11 | 10 | 12 | 13 | 15 | 15 | 20 |
| 2014 | 27 | 18 | 15 | 9 | 17 | 11 | 16 | 17 | 14 | 19 |
| 2015 | 25 | 13 | 16 | 8 | 10 | 11 | 10 | 15 | 18 | 27 |
| 2016 | 22 | 14 | 12 | 15 | 10 | 17 | 13 | 16 | 15 | 18 |
| 2017 | 26 | 15 | 13 | 9 | 11 | 11 | 12 | 14 | 17 | 15 |
| 2018 | 24 | 11 | 11 | 7 | 19 | 14 | 15 | 16 | 15 | 18 |

**Table 9.** *Cont.*

| Year | Norwegian Sea | Barents Sea | Kara Sea | Laptev Sea | East Siberian Sea | Chukchi Sea | Beaufort Sea | Northwest Passage | Baffin Bay | Greenland Sea |
|------|------|------|------|------|------|------|------|------|------|------|
| 2019 | 23 | 14 | 12 | 8 | 8 | 12 | 10 | 13 | 13 | 18 |
| 2020 | 20 | 12 | 15 | 9 | 9 | 11 | 16 | 14 | 14 | 19 |
| 2021 | 14 | 13 | 12 | 12 | 9 | 12 | 11 | 15 | 14 | 14 |

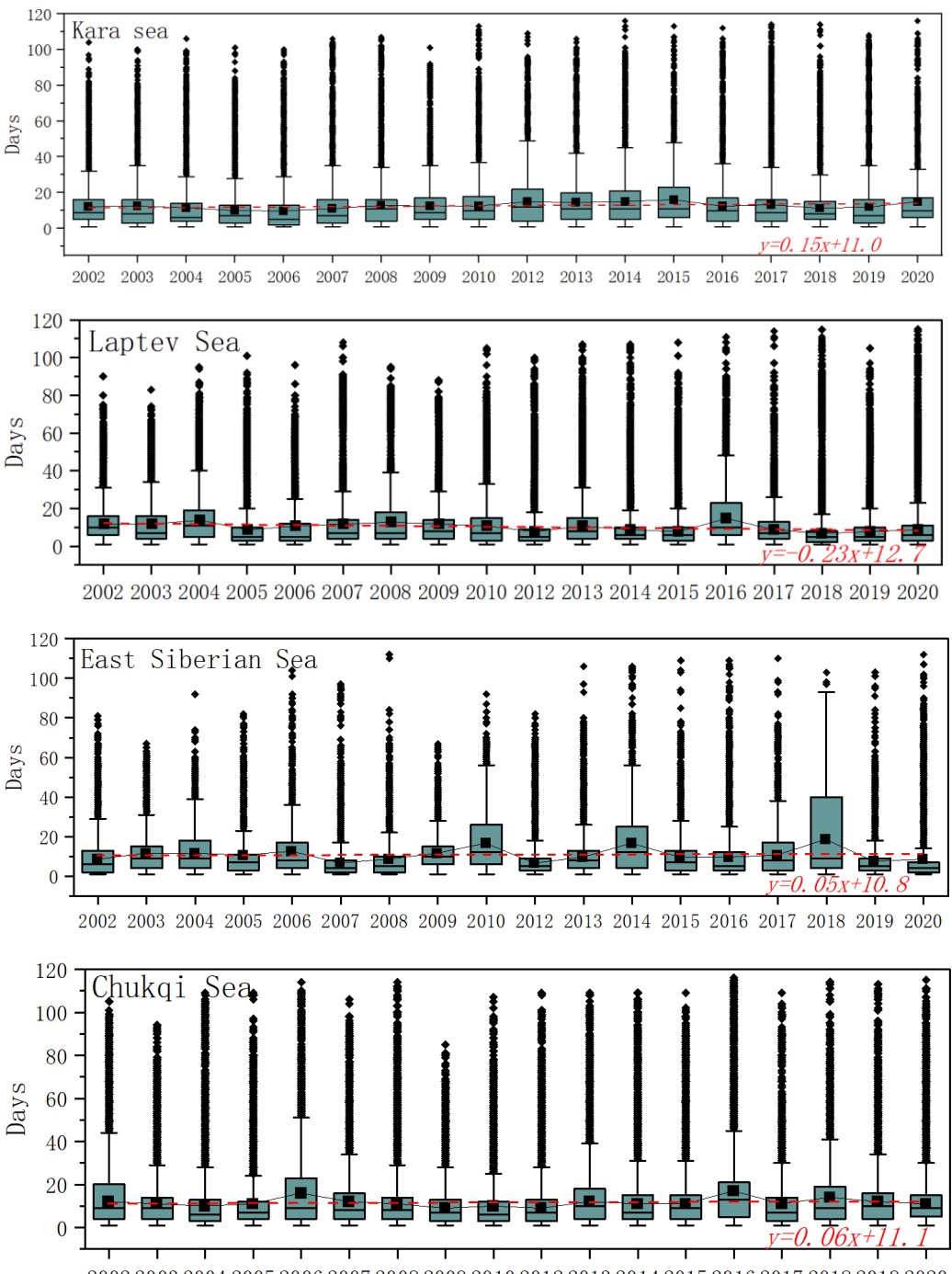

**Figure 15.** Number of days from freezing up to complete freezing of sea ice in Kara Sea, Laptev Sea, East Siberian Sea, and Chukchi Sea from 2002 to 2020. The standard errors were 0.07, 0.09, 0.15, and 0.09, respectively, and passed the significance test ($p < 0.05$).

### 5.2.3. Early Melting and Late Freezing

Arctic sea ice melts earlier and freezes later [55]. During our study period, the trend of early melting of Arctic sea ice is very prominent in some sea areas (Figure 16), but the overall trend of early melting is not significant. The Kara Sea showed the most obvious trend of early melting, with the trend of the beginning of melting and complete melting being −1.22 d/year and −1.20 d/year, respectively. The Laptev Sea followed, with an early trend of −1.17 d/year for the beginning of melting and −0.99 d/year for the complete melting. In the Barents Sea, the advance trend of complete melting was −1.02 d/year and −0.76 d/year, respectively, faster than that of the beginning of melting. The East Siberian Sea also showed an obvious trend of early melting, with a trend of −0.65 d/year for the beginning of melting and −0.54 d/year for the complete melting. In addition, the Norwegian Sea also showed a slightly earlier trend, with the onset and complete melting time of −0.20 d/year and −0.22 d/year, respectively. It is worth noting that the Northwest Passage also showed an earlier trend of −0.55 d/year, but the time to complete melting was not significantly earlier. The trend of melting in the remaining Arctic marginal waters was not significant.

There is a significant delay of freezing in the Arctic as a whole, with a delayed trend of 0.65 d/year for the beginning of freezing and 0.69 d/year for the complete freezing, respectively. The trend of freezing delay in the Arctic is more significant than that of melting. The Kara Sea has the strongest delayed freezing trend in the Arctic, which is +1.92 d/year and +1.77 d/year, respectively (Figure 17). The Laptev Sea, which is adjacent to the Kara Sea, has the second most significant trend, with a delayed trend of +1.85 d/year onset and +1.61 d/year complete freezing. The Barents Sea followed, with a delayed trend of +1.38 d/year for onset and +1.64 d/year for complete freezing, respectively. In the East Siberian Sea and Chukchi Sea, the delay trend of sea ice freezing onset was +1.07 d/year and +1.16 d/year, respectively. In addition, the Beaufort Sea also showed a significantly delayed trend of sea ice freezing, with a delayed trend of +0.79 d/year and +0.78 d/year for the beginning of freezing and complete freezing time, respectively. The accelerated decline in SIC is associated with warmer water temperatures in the Pacific and rising heat flux in the Chukchi and Beaufort Seas, which delay the formation of sea ice [56]. The number of thawing days and freezing days in the Norwegian Sea and Kara Sea showed a significant increasing trend. The Northeast Passage, including the Barents Sea, Kara Sea, Laptev Sea, and East Siberian Sea, showed the largest trend of early and delayed Arctic melting and freezing among the Arctic marginal seas. This trend shows that in recent years, the earlier melting time and the later freezing time of the Northeast Passage expanded the navigation window and made the navigation time longer, which promoted the international economic exchanges using the Arctic route and brought great convenience to the navigation of the Arctic Passage.

There is also an interesting phenomenon that there is a clear circular band in space in the annual Arctic freezing days, and the number of freezing days in this range is significantly greater than on either side of the circular band, indicating that a long time passes between the beginning of freezing and complete freezing (Figure 14). In the range west of 120°E and west of 60°W, this band appears at approximately 75°N, and the other side of the band appears at approximately 82°N. This may be related to the different degrees of heat transport from the North Atlantic and the Pacific to the Arctic [57]. However, rising SST in both the North Atlantic and North Pacific will cause Arctic sea ice to melt faster [58]. On the side near the Atlantic, the band is closer to the North Pole, possibly due to the influence of warm currents that push it toward the North Pole. We suspect that the reason for this is that outside the band, the low latitudes of the Arctic are heavily influenced by the exchange of fresh water from land and heat from the Atlantic and Pacific oceans, and during freezing periods, these regional sources do not provide enough heat to allow the sea ice to completely freeze in a shorter period of time. The inner part of the band, near the North Pole, has a great deal of multi-year ice and less extensive melting, and if it does melt, it can quickly refreeze, so it exhibits a shorter freezing process. This band is located in

an important position between the closed Arctic and the open ocean and represents the difference between the two Arctic environments.

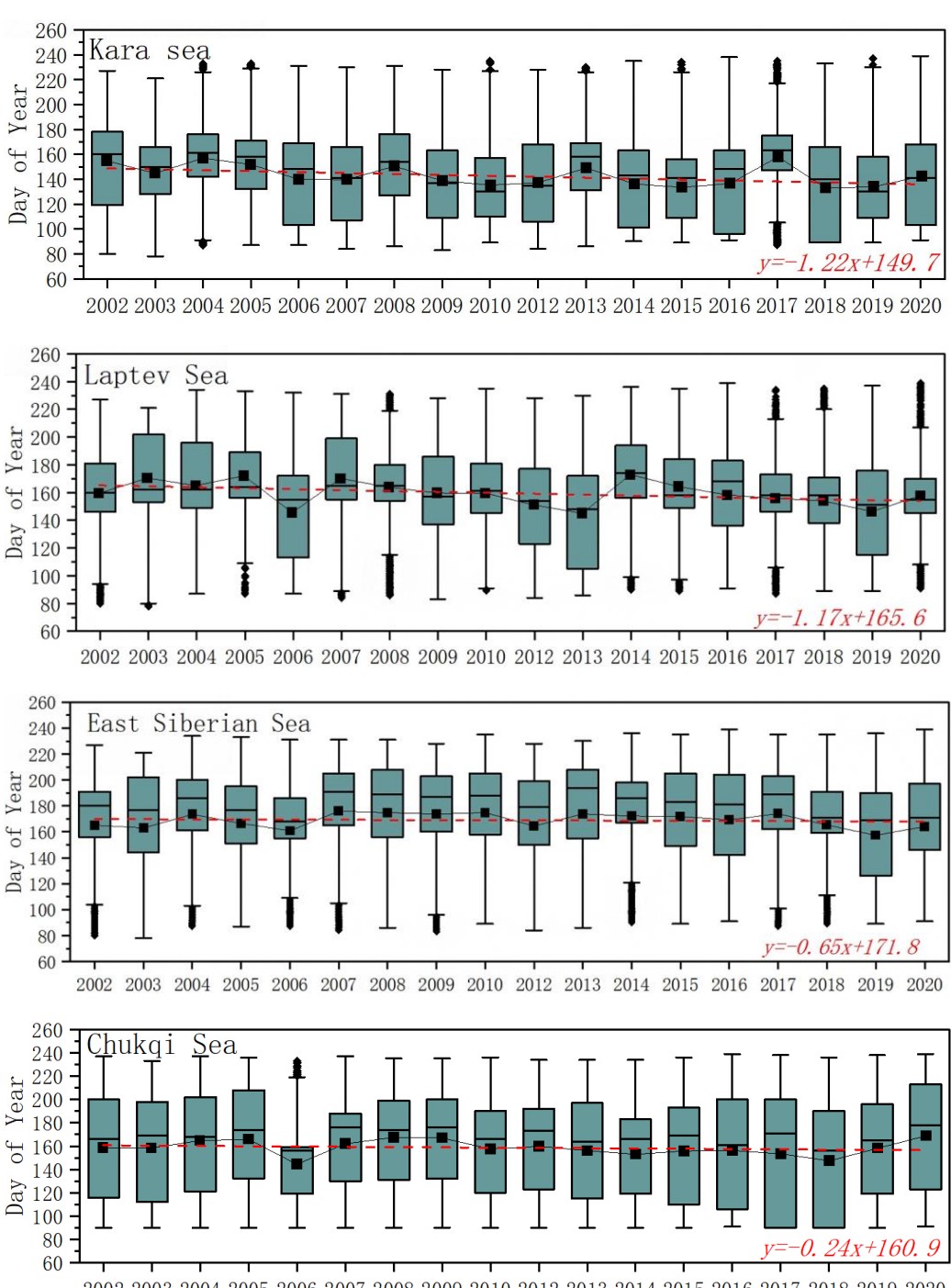

**Figure 16.** Temporal distribution of sea ice break-up dates in the Kara Sea, Laptev Sea, East Siberian Sea, and Chukchi Sea in the Arctic from 2002 to 2020, which shows an earlier break-up for all selected sea areas. The standard errors were 0.35, 0.38, 0.26, and 0.30, respectively, and passed the significance test ($p < 0.05$).

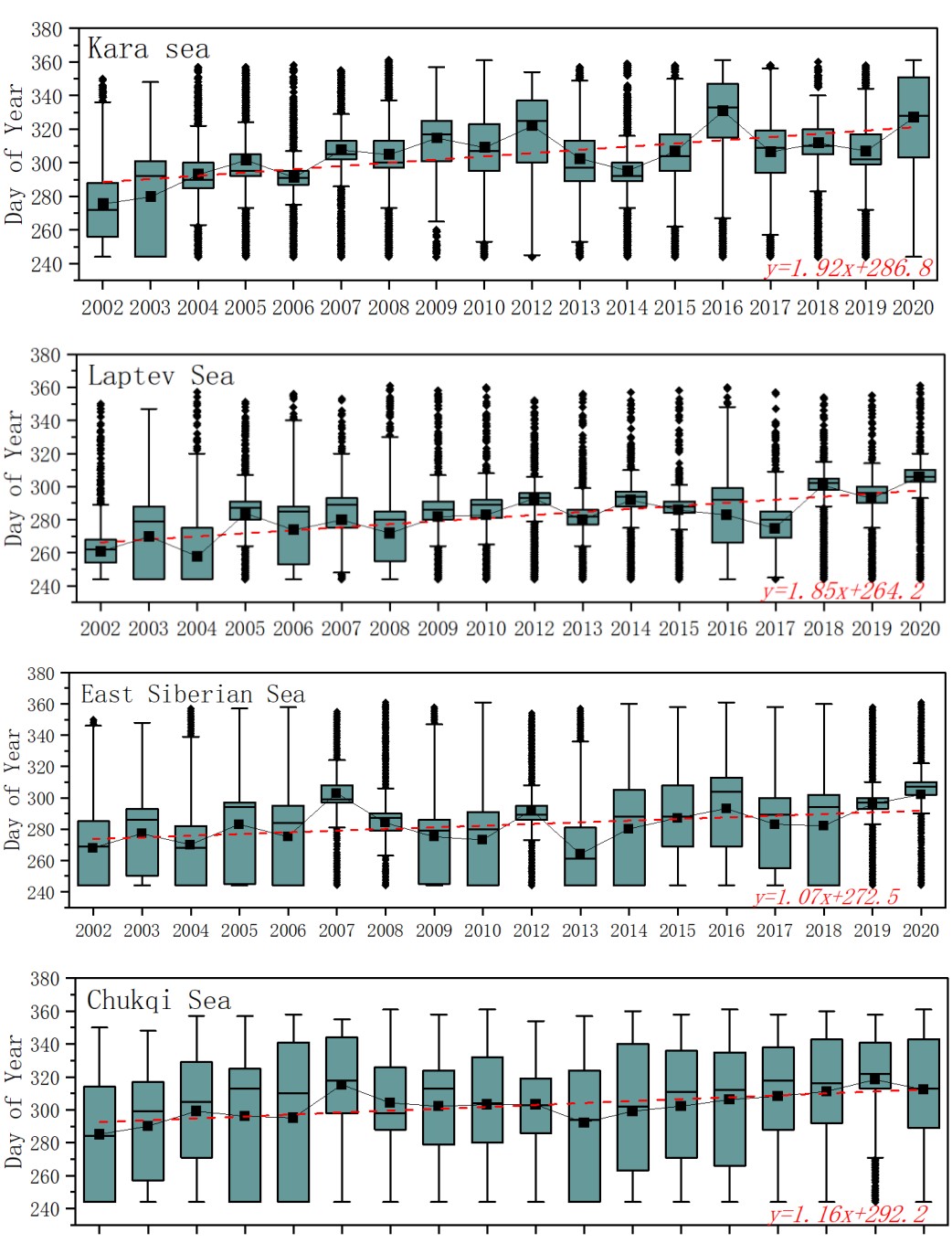

**Figure 17.** Temporal series analysis of sea ice freeze-up dates in the Kara Sea, Laptev Sea, East Siberian Sea, and Chukchi Sea in the Arctic from 2002 to 2020, which shows an obviously postponed trend for freeze-up for all the selected seas. The standard errors were 0.48, 0.36, 0.45, and 0.28, respectively, and passed the significance test ($p < 0.05$).

## 6. Discussion

The difference in the Arctic marginal seas is greatly influenced by the North Atlantic current, which carries warm water northward [8] through the Bering Strait, providing an important source of fresh water and heat for the Arctic Ocean [59,60]. Atlantic Water (AW) and halocline waters flow along the Siberian shelf of the Laptev Sea as a triple-core current: The conventional Fram Strait Branch (FSB) and Barents Sea Branch (BSB) and the Arctic Shelf Break Branch (ASBB), which are important for the sea ice, heat balance, and ocean circulation in the Arctic Ocean [61]. AW sufficiently carries heat to the ocean

surface, thinning Arctic sea ice and causing it to gradually retreat [62], and even melting several meters of sea ice in a few years [63,64]. This energy exchange occurs in atmosphere–ice–ocean interactions. The atmosphere and ocean interact via the interface at the sea surface, and the SST and sea ice are therefore crucial elements for atmosphere–ice–ocean interactions [65].

Heat transport through the Barents Sea is the primary source of internal variation in winter Arctic sea ice, affecting the entire Arctic Ocean [66]. The loss of sea ice in the central Arctic is primarily influenced by heat transported to the Arctic Ocean through the Fram Strait, which causes the sea ice at its bottom to melt [67]. The delay in freezing in the Laptev Sea caused by unusually prolonged warmth over northern Russia and the intrusion of the Atlantic current could have an Arctic chain reaction. On the other hand, the influence of freshwater rivers flowing into the Arctic on SST and the melting and freezing process of sea ice cannot be ignored. In the coastal zones where rivers such as the Ob River, Yenisey River, and Lena River flow into the Arctic, the SST increases and the SIC decreases (Figure 18). Increased runoff can affect various processes related to heat and salt diffusion, ultimately leading to an increase in advection heat and salt flowing into the Arctic [68]. Net sea-ice melt (~109–158 km$^3$) is only present in years with high river water budgets [69]. More river runoff has been linked to more summer melt, as well as earlier freezing [70].

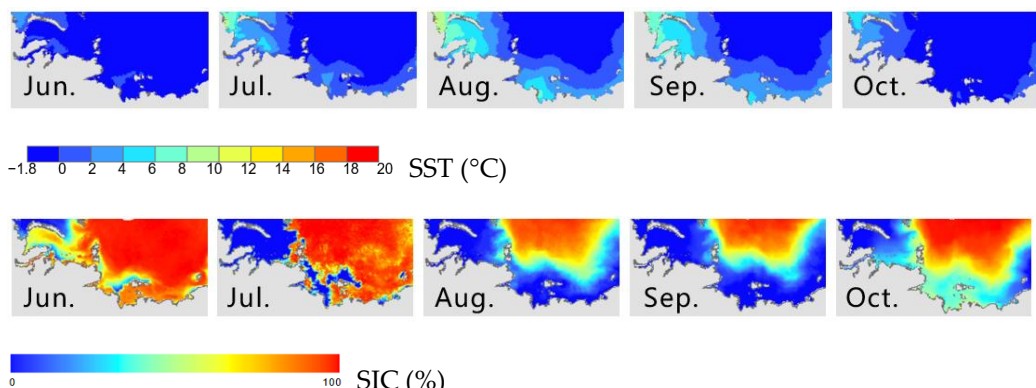

**Figure 18.** Monthly mean of SST and SIC in Northern Siberia from June to October 2002 to 2021.

Changes in SST are not the only factor delaying the freezing of sea ice. Climate change will also push more warm Atlantic water towards the Arctic, affecting ice formation, which may accelerate the decline of Arctic sea ice. Surface circulation is primarily the transpolar drift across the Arctic Basin from Eastern Siberia and the Laptev Sea to the Fram Strait and the anticyclonic Beaufort Gyre in the Canada Basin [71]. The Arctic circulation is not only closely related to the Arctic Oscillation [72] but also plays an important role in ocean warming and sea ice decline. Sea ice motion with the anticyclonic flow of the Beaufort Gyre and Transpolar Drift Stream is evident. In addition, surface freshwater from six primary Arctic rivers is drawn toward the center of the Canadian Basin by the anticyclonic winds of the Beaufort High, ensuring the maintenance of the Arctic's strong halocline stratification, which affects the growth of sea ice as it grows and moves [73].

The temperature response to sea ice loss may be temporally inconsistent with sea ice extent change [74], and the greatest impact of sea ice loss on the Arctic climate will occur from mid-autumn to late autumn, rather than in late summer when the sea ice extent is minimum. Seasonal Arctic sea ice usually goes through a cycle of melting and refreezing, with annual changes in SIC. However, for multi-year ice, even in summer, the sea ice intensity remains high, staying close to 100% [55]. The large area of multi-year ice near the North Pole is covered, and even if it melts, it only forms melt pools on the ice, making it difficult to accurately monitor its status. Due to a lack of observational data, there is not yet a comprehensive understanding of the effects of the thin ice freezing process, which is complicated by the formation of ice slime containing seawater exposed to the

air [75]. Ice melting is influenced by precipitation [76,77], circulation [78], snow cover, water accumulation or drainage, ice thickness, wind direction or strength, air temperature, water depth [79], etc. The length of the melt season has been increasing in the Arctic at a rate of 5 days per decade [80]. During the warmest time of the year, the sea ice begins to melt, and liquid water begins to pool on its surface. Sea ice thickness also contributes to the length of the process [81].

## 7. Conclusions

Based on the daily SST dataset provided by NCEI and the daily SIC data of the University of Bremen, the changes in SST and SIC in the Arctic from 2002 to 2021 and their correlation were analyzed, and the changes in sea ice melting and freezing in the Arctic in the past two decades were discussed. The conclusions we drew are briefly discussed.

The highest and lowest monthly mean Arctic SST values occur in August and March, respectively, while the SIC is in March and September. The annual variation of the Arctic SST shows that the overall warming trend is not obvious. Compared with other marginal sea areas, only the Kara Sea, Barents Sea, and Laptev Sea showed a significant increase in SST, with a trend of 0.068 °C/year, 0.052 °C/year, and 0.044 °C/year, respectively. The annual mean of Arctic SIC shows a significant downward trend of approximately −0.31%/year. The trends of SST and SIC in autumn are the most significant, which are +0.01 °C/year and −0.45%/year, respectively.

There is a significant negative correlation between SST and SIC in the study area and the correlation coefficient is −0.82, but there are significant differences in different seas. The correlation is low at the entrance of the North Atlantic Ocean and high in the Northeast Passage area. The change in sea ice is influenced by more factors, primarily by the inflow of warm water from the North Atlantic and Pacific oceans. The Northeast Passage is relatively closed and has a more stable environment, with a stronger correlation between SST and SIC. It should be noted that partial sea ice in winter is completely covered, and the corresponding sea surface temperature data are expressed as nearly −1.8 °C, while partial sea area is ice-free in summer. As a result, the relationship between SST and SIC becomes complicated and difficult to quantify with accurate data, which is a problem to be solved in the follow-up work.

The sea ice break-up occurs on DoY 143 and freeze-up occurs on DoY 296 in the Arctic. The sea ice break-up occurs first in the Norwegian Sea and last in the East Siberian Sea. The duration of rapid melting and freezing is 27 and 14 days, respectively. However, the rapid melting and freezing processes do not significantly increase or decrease. The sea ice break-up is advanced and the freeze-up is delayed, and the trend toward earlier melting is smaller than the trend toward later freezing. The Northeast Passage is the sea area with the most significant advance of sea ice melting and the Kara Sea has the strongest trend of −1.22 d/year, followed by the Laptev Sea ay −1.17 d/year. The delay trend of sea ice freezing was most significant in the Kara Sea with +1.75 d/year, followed by the Laptev Sea with +1.70 d/year. However, there is a dynamic process of ice to take into account, as the wind and the ocean cause the sea ice to drift. Ice undergoes deformation and thermal processes that cause it to grow, melt, and break up.

By analyzing the temporal and spatial characteristics and variation trends of SST and SIC in the Arctic, as well as the changes in the rapid melting and freezing of sea ice in the Arctic, it is more helpful to understand the environmental changes in the Arctic, especially the significant changes in the Northeast Passage compared with other marginal sea areas, which is of great significance to shipping in the Arctic. However, this paper does not consider the influence of other factors such as ocean currents, and further research is needed.

**Author Contributions:** Conceptualization, Y.Q. and M.Y.; methodology, M.Y. and Y.Q.; project administration, Y.Q. and J.C.; supervision, Y.Q.; visualization, Z.J.; writing—original draft, M.Y.; writing—review and editing, Y.Q., M.Y., L.H., M.C. and B.C. All authors have read and agreed to the published version of the manuscript.

**Funding:** This research was funded by the National Key Research and Development Program of China (No. 2019YFE0105700 and No. 2017YFE0111700), the Innovative Research Program of the International Research Center of Big Data for Sustainable Development Goals (No. CBAS2022IRP08), the Strategic Priority Research Program of the Chinese Academy of Sciences (No. XDA19070201) and the International Partnership Program of Chinese Academy of Sciences "Remote Sensing and Modeling of the Snow and Ice Physical Process (RSMSIP, No. 313GJHZ2022054MI)".

**Data Availability Statement:** The sea surface temperature data OISST Version 2.1 are from the National Centers for Environmental Information (NCEI, https://www.ncei.noaa.gov/, accessed on 16 June 2022), the sea ice concentration data are from the University of Bremen (https://seaice.uni-bremen.de/data, accessed on 16 June 2022), and the sea ice extent data are from the U.S. National Ice Center (USNIC, https://usicecenter.gov/Products/, accessed on 3 October 2022).

**Acknowledgments:** The authors would like to thank the National Centers for Environmental Information, the University of Bremen, and the U.S. National Ice Center. Moreover, we thank the Digital Belt and Road (DBAR) program Working Group on the High Mountain and Cold Regions (HiMAC) and the Digital Arctic Shipping-New Data Products and Visualisation Services (DARC-SERV).

**Conflicts of Interest:** The authors declare no conflict of interest.

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
