# Peer review of "Changes in Sea Surface Temperature and Sea Ice Concentration in the Arctic Ocean over the Past Two Decades"

_remotesensing, doi:10.3390/rs15041095_

Round 1

Reviewer 1 Report

This study investigates the changes of SST and SIC in the Arctic in the past twenty years using data products. This research is very interesting and important. The manuscript is also well organized. But I have several concerns which the authors should clarify. I recommend this manuscript to be accepted after major revision.

Comments:

1) There are limited historical SST observations in the Arctic. Please justify the validity of SST data for this study.

2) This research investigates the relationship between the ice and ocean. However, previous study reveals on interannual time scales, the sea ice in the Arctic is controlled by the atmosphere. Therefore, when the authors draw the conclusions between ocean and sea ice, please add the discussion of the role of atmosphere and present a logical story.

3) The authors show the correlation between SIC and SST for different regions in the Arctic. How about the correlations on different time scales? How about the annual correlations in different seasons? As correlation can’t present causality, how did the authors investigate the response of SST (Section 5.2) and influences of SST?

Author Response

Dear reviewer 1:

Thank you for your comments concerning our manuscript. 

Reviewer 2 Report

This paper investigates the evolution of two variables: the sea surface temperature, and the sea ice concentration in the Arctic ocean. Eleven regions are considered above 60º N, the Central Arctic and ten seas that round it. The period investigated extended from 2002 to 2021. Linear fits are used to calculate and compare the trends in the regions suggested. The research separates between ice breaking and complete melting, and ice freezing and complete freezing. A key point of the paper is the spatial representation of the processes investigated, which is presented every year. Varied tables show the results for the ten seas around the Arctic. Moreover, the correlation between the two variables above cited is presented, and the melting and freezing dates are investigated. Finally, varied references are considered along the text. Consequently, the paper is quite complete and could be published in Remote Sensing after the introduction of the following minor changes.

Certain quantities are presented with their uncertainty. However, the authors should indicate if these uncertainties are the standard deviations. Moreover, other quantities, such as certain trends are presented without uncertainties along the text. The authors should present the trends with their uncertainties (Section 5.2.3) and indicate if these trends are statistically significant.

Figure 1. Ten seas are around the Arctic. However, their extension is quite varied. Since the seas are determined by their pixels, the ratio against the total surface can indicate the weight of each sea.

Section 3.1. The sea surface temperature is provided with a 0.25º resolution. However, the monthly average is calculated by pixels, L. 175. The authors should indicate the procedure to interpolate the data from the original database.

Section 3.2, L. 182. The names of the independent variable, such as time, and dependent variable should be introduced.

L. 312-315. Four points are selected. The authors should indicate the reason for such a selection.

Figure 6. Certain pixels present positive correlation, i.e., an increase of the sea surface temperature is accompanied by an increase of the sea ice concentration. The authors should explain this result.

Minor remarks

L. 64. The decimal place of the last figure should be the same for the quantity and for its uncertainty. Moreover, the uncertainty is similar or even greater than the quantity.

L. 188. The range for the correlation coefficient is (-1,1).

Titles of the sections 4.1 and 4.2 should begin with capital.

Figure 2. The first row is incomplete. The bottom of the figure is cut. The colour scale is missing.

Figure 3. The first row is incomplete. The bottom of the figure is cut.

L. 388. The comma at the end of the sentence should be supressed.

L. 589. Replace comma after [63] by stop.

Author Response

Dear reviewer 2:

Thank you for your comments concerning our manuscript. 

Reviewer 3 Report

The article is interesting and is part of the worldwide trend regarding temperature and sea ice research. I recommend the article for publication after taking into account the following comments.

The abstract: is too broad and should refer to the most relevant things in the article. In addition, its length is inconsistent with MDPI guidelines.

1. Introduction.

I did not find a clearly defined purpose of the paper. The authors present a broad literature and, therefore, what is their contribution to the state of existing knowledge supposed to be? This should be clearly and precisely stated.

The period of analysis is 2002-2021. It is standard in long-term studies of climate and hydrological changes to use a period of 30 years (minimum). Please clarify this point.

5. Results and Analysis

Figure 2: What do the different colors mean?

6. Discussion and Conclusions

In the discussion section, I do not find a broader interpretation of the results obtained.

The authors omit the issues of the supply of water of different temperature and ice by large rivers flowing into the Arctic ocean. This is especially relevant for the coastal zone.

It would be interesting to show the circulation situation during the analyzed period, which may have important implications for ice drift.

Distribution of sea currents. Admittedly, the authors state that this is a topic for further research (detailed, I assume), but a reference to the general conditions of the distribution of currents is most welcome.

The reference to the Arctic Shipping Route (key words) is not confirmed in the text. The authors do not analyze this issue more extensively in the context of the results obtained.

Author Response

Dear reviewer 3:

Thank you for your comments concerning our manuscript. 

Round 2

Reviewer 1 Report

The authors have successfully addressed my questions.

Author Response

Dear Reviewer,

Thank you for your comments concerning our manuscript.  We uploaded a new revised manuscript.